# Classical Planning with LLM-Generated Heuristics: Challenging the State of the Art with Python Code

**Augusto B. Corrêa**
University of Oxford
United Kingdom

**André G. Pereira**
Federal University of Rio Grande do Sul
Brazil

**Jendrik Seipp**
Linköping University
Sweden

## Abstract

In recent years, large language models (LLMs) have shown remarkable performance in many problems. However, they fail to plan reliably. Specialized attempts to improve their planning capabilities still produce incorrect plans and fail to generalize to larger tasks. Furthermore, LLMs designed for explicit "reasoning" fail to compete with automated planners while increasing computational costs, which reduces one of the advantages of using LLMs. In this paper, we show how to use LLMs to always generate correct plans, even for out-of-distribution tasks of increasing size. For a given planning domain, we ask an LLM to generate several domain-dependent heuristic functions in the form of Python code, evaluate them on a set of training tasks with a greedy best-first search, and choose the best one. The resulting LLM-generated heuristic functions solve substantially more unseen out-of-distribution test tasks than end-to-end LLM planning, particularly for non-reasoning LLMs. Moreover, they also solve many more tasks than state-of-the-art domain-independent heuristics for classical planning, and are competitive with the strongest learning algorithm for domain-dependent planning. These results are impressive given that our implementation is based on a Python planner and the baselines all build upon highly optimized C++ code. In some domains, the LLM-generated heuristics expand fewer states than the baselines, showing that they are not only efficiently computable but also more informative than the state-of-the-art heuristics. Overall, our results show that sampling a set of planning heuristic functions can significantly improve the planning capabilities of LLMs.

## 1 Introduction

Classical planning is a fundamental problem in Artificial Intelligence (AI), with applications ranging from robotics to computational chemistry [25]. Given the *initial state* of the world, a description of the *goal*, and a set of deterministic *actions* that can be executed in a fully-observable environment, the task is to find a sequence of actions transforming the initial state into a state that satisfies the goal. Nowadays, most classical *planners* rely on *heuristic search* algorithms to find plans [6, 39, 35, 55, 45, 75, 65]. The efficiency of these planners depends on the quality of the *heuristic functions* that estimate the cost of reaching the goal from a given state. Traditionally, these heuristics have been either *domain-independent*, offering generality at the expense of accuracy; *manually crafted* for specific domains, requiring significant human effort and expertise; or *learned on a per-domain basis*, incurring costs for training a new heuristic whenever we want to use a new domain.

In this work, we propose a new way of producing heuristics: we use LLMs to automatically generate domain-dependent heuristic functions for classical planning. Our hypothesis is that LLMs, given sufficient context and examples, can generate heuristic functions that outperform generic domain-independent heuristics.

39th Conference on Neural Information Processing Systems (NeurIPS 2025).

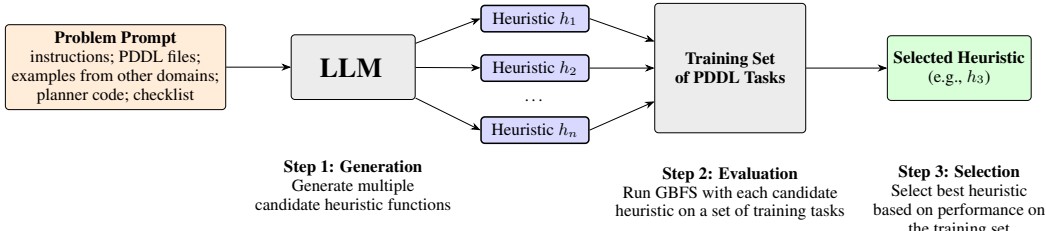

Figure 1: Our pipeline for generating domain-dependent heuristics with LLMs: we prompt the LLM $n$ times to generate $n$ candidate heuristics, evaluate each of these heuristics on a set of training tasks and choose the strongest one.

Our overall pipeline is much simpler than previous work: we simply pass to an LLM the domain description, example planning tasks, example domain-dependent heuristics for other domains, and the relevant planner code. Then we request that the LLM generates a heuristic for the given domain. We specifically request that the LLM *generates the code*, in Python, to compute the heuristic. We execute the same prompt $n$ times to obtain a pool of $n$ candidate heuristics, evaluate each of them on a training set, and select the best one. Figure 1 shows the overall pipeline. This discards the necessity of a back-and-forth communication between the planner and the LLM, making the overall procedure straightforward. Our approach generates a constant number of heuristics *per domain*, and then uses the selected heuristic for any new task of this domain. This drastically reduces the costs for LLM inference compared to invoking an LLM for each task in a domain.

We implement our pipeline on top of Pyperplan [1] as a proof of concept, and then evaluate the generated heuristics on the domains of the Learning Track of the International Planning Competition (IPC) 2023 [73]. The LLM-generated heuristic functions solve significantly more tasks than LLMs prompted to generate plans end-to-end, and this difference is especially pronounced for non-reasoning LLMs. The heuristics also outperform state-of-the-art heuristics, such as $h^{FF}$ [39] in terms of solved tasks and are competitive in the number of required state expansions. Although Pyperplan is much slower than state-of-the-art planners [35, 45, 63] due to its Python implementation, our method still outperforms $h^{FF}$, even when using the standard C++ implementation available in Fast Downward [35], as well as the strongest learning-based domain-dependent planner [11], which is also built on Fast Downward. This is an impressive result, as this implementation is a cornerstone of most of the state-of-the-art planners in the literature.

## 2 Background

We consider *classical planning*, where a single agent applies deterministic actions in a fully-observable, discrete environment. Classical planning tasks are usually described using the Planning Domain Definition Language (PDDL) [47, 33]. To understand our work, an informal description of a fragment of PDDL is sufficient. We introduce it alongside examples from a simple Logistics domain.

A PDDL task consists of a set of objects (e.g., representing vehicles, packages, locations); a set of predicates representing relations between these objects (e.g., using the `at` predicate, the *ground atom* (`at car1 city2`) represents that object `car1` is at `city2`); a set of actions that change the relations (e.g., driving a car changes its location); an initial state which is a set of ground atoms (e.g., where all packages and vehicles are initially and which locations are connected) and a goal description that lists the ground atoms that must hold at the end of a plan (e.g., the desired locations of all packages). Applying an action changes the current state of the environment by removing or adding ground atoms. PDDL tasks are commonly separated into a *domain* and an *task* part, where the domain part holds the common actions and predicates, while the task part describes the specific objects, initial state and the goal. The two parts are typically represented as two separate files.

The objective of a *planner* is to find a sequence of actions, called a *plan*, that leads from the initial state to a state where all goal atoms hold. Most planners nowadays use *state-space search* to find a plan. The search is usually guided by a *heuristic* function $h$ which maps each state $s$ to a value $h(s) \in \mathbb{R}_0^+ \cup \infty$ that estimates the cost of reaching a goal state from $s$ [51]. In heuristic search algorithms—such as A*[32], weighted-A*[52], and greedy best-first search [19]—this heuristic

guides the search towards promising states and thereby reduces the search effort. The performance of a planner is heavily influenced by the accuracy and computational efficiency of the heuristic function.

In our work, we assume that all actions have cost one, and we consider *satisficing planning*, where any plan is acceptable, irrespective of its length or cost. We focus on planners that use greedy best-first search (GBFS). While there are lots of search improvements that one could evaluate on top of GBFS [e.g., 39, 45, 54, 57], we limit ourselves to "pure" GBFS planners as this is the most commonly used version in the classical planning literature [e.g., 18, 22, 38].

## 3    Using LLMs to Generate Heuristics for Classical Planning

In our pipeline, we give as input to an LLM the PDDL description of our target domain together with some additional information (described below). Then we ask the LLM for a domain-dependent heuristic function for the given domain, implemented in Python, which we then inject into the Pyperplan planner [1]. We choose Python because LLMs generate correct code for Python more often than for other languages [44], and because code injection is simple in Python.

We send $n$ identical requests to the LLM with the same prompt, collect all returned heuristic functions $h_1, \ldots, h_n$, and then evaluate them on the training tasks. Then, we automatically select the best heuristic $h_{\text{best}} \in \{h_1, \ldots, h_n\}$ and use it in the test set evaluation (see below for details on this selection). Figure 1 shows the graphical representation of our method.

When used for planning, LLMs often produce incorrect plans [79, 80]. In contrast, our pipeline ensures that all found plans are correct: the greedy best-first search algorithm only produces correct plans, and the heuristic created by the LLM only influences how efficiently such a solution is found.

**Prompt**    Our prompt instructs the LLM to generate a domain-dependent heuristic for a given domain $D$ and provides some advice, such as that the heuristic should minimize the number of expanded states and balance accuracy with computational efficiency. It then includes the following file contents:

1. the PDDL domain file of domain $D$

2. the smallest and the largest PDDL tasks of domain $D$ in the training set

3. for each of the two example domains Gripper and Logistics [47]: the PDDL domain file, a task file and a domain-dependent heuristic implemented in Pyperplan

4. an example of how a state of domain $D$ is represented in Pyperplan

5. an example of how the static information of domain $D$ is represented in Pyperplan

6. the Python code from Pyperplan for representing a planning task and an action

7. a checklist of common pitfalls

Items 1 and 2 provide the context about the domain $D$ that we are interested in. Item 3 illustrates the Python implementation of domain-dependent heuristics using Pyperplan's interface. For Gripper, we provide a Python function computing the perfect heuristic as input, while for Logistics, we encode the simple "single visit and load/unload counting heuristic" by Paul et al. [50]. These functions show the LLM what a heuristic could do and illustrate how to manipulate the available data. Items 4–6 give more context about Pyperplan. Last, the checklist consists of tips based on our own observations of LLM responses. Appendix A shows the complete prompt used for the Blocksworld domain [47].

**Heuristic Function Selection**    We prompt the model $n$ times with the input above to generate $n$ heuristics. For each task in the training set, we then run the $n$ heuristics within a GBFS for at most five minutes. Then, we select the heuristic that solves the largest number of tasks from the training set. If there is a tie, we choose the one minimizing the accumulated *agile score* [1] over the training set. We use the term *training set* to follow IPC terminology. However, our approach does not involve any training. We merely use this set of tasks to evaluate the generated heuristics and select the best one.

---

[1]The agile score is a common metric from IPCs and awards heuristics that lead to finding plans quickly. If the search needs less than 1 second, then the score is 1. If the search runs out of time (in our case, 300 seconds) the score is 0. Intermediate values are interpolated with the logarithmic function $1 - \frac{\log(t)}{\log(300)}$, where $t$ is the run time (in seconds) of the search. The accumulated score is the sum over all training tasks.

Table 1: Size of training and testing tasks for each domain based on their main parameters. Parameters by domain: $n$ blocks in Blocksworld, $c$ children in Childsnack, $t$ tiles in Floortile, $p$ passengers in Miconic, $r$ rovers in Rovers, $b$ boxes in Sokoban, $s$ spanners in Spanner, and $v$ vehicles in Transport.

| Domain | Training | Testing |
|---|---|---|
| Blocksworld | $n \in [2, 29]$ | $n \in [5, 488]$ |
| Childsnack | $c \in [1, 10]$ | $c \in [4, 292]$ |
| Floortile | $t \in [2, 30]$ | $t \in [12, 980]$ |
| Miconic | $p \in [1, 10]$ | $p \in [1, 485]$ |
| Rovers | $r \in [1, 4]$ | $r \in [1, 30]$ |
| Sokoban | $b \in [1, 4]$ | $b \in [1, 79]$ |
| Spanner | $s \in [1, 10]$ | $n \in [1, 487]$ |
| Transport | $v \in [1, 7]$ | $n \in [3, 50]$ |

## 4  Experimental Results

For running our experiments, we use Downward Lab [64] on AMD EPYC 7742 processors running at 2.25 GHz. As mentioned, we use Pyperplan [1] for all our configurations. This allows us to evaluate the different heuristics (domain-independent and LLM-generated ones) in a single framework. We use PyPy 7.3.9 to run Pyperplan, as it proved to be slightly faster than CPython. The source code and experimental data are publicly available online [14].

In the training phase, we limit each run to 5 minutes and 8 GiB. In the testing phase, each run is limited to 30 minutes and 8 GiB, following recent International Planning Competitions [73].[2] To diversify the pool of generated heuristics, we increase the temperature parameter of the models to 1.0 [7].

We use the domains and training/test tasks from the IPC 2023 Learning Track to generate and evaluate heuristics. However, since Pyperplan does not support two of these ten domains (Ferry and Satellite), we exclude them from our experiments. The resulting test set has 90 tasks for each of the 8 domains. The distribution of tasks in the training and test sets differs a lot: the test tasks are generally much larger than the training ones. Table 1 shows the distributions of parameters. In addition to size differences, tasks may also vary in structure. For example, Sokoban mazes can be arranged in different layouts. The full details about the task sets are available online [62].

We also include experiments on novel domains that were not seen during the training of the LLMs. This helps us to identify whether our method is robust to new domains, and also addresses the potential concern that the LLMs are retrieving memorized heuristics instead of generating them by reasoning about the domain description.

**Generating Heuristics**  To generate the heuristics, we use two families of LLMs: Gemini [27, 28], with the models Gemini 2.0 Flash (stable release 001) and Gemini 2.0 Flash Thinking (version 01-21); and DeepSeek [16, 17], with the models DeepSeek V3 (version 0324), DeepSeek R1 Distill Qwen 14B, and DeepSeek R1. We include the distilled version to Qwen 14B [3] to evaluate the impact of smaller models in our pipeline. We had free API access to Gemini 2.0 models and ran R1 Distill locally. The total API costs for generating all V3 and R1 heuristics were $0.25 and $6.12 (USD).

We prompt the LLM $n$ times and receive $n$ different heuristic functions. But how large should $n$ be? We ran an experiment with Gemini 2.0 Flash to evaluate this. For all domains, the biggest increase in average *coverage* (i.e., number of solved tasks) results from going from 1 to 5 heuristics, and after that, we see diminishing returns. In six of the eight domains, using $n = 25$ is enough to consistently find heuristics solving the entire training set. In two domains, Childsnack and Floortile, coverage increased for $n > 25$, but only slightly. Due to these results, we set $n = 25$ for all experiments below.

---

[2]We used a disjoint set of ten domains from the Autoscale benchmark set [76] for exploratory experiments while developing our pipeline. This split allowed us to test different prompts, hyperparameters, and models without the risk of overfitting to the IPC 2023 benchmark set or causing some LLM APIs to cache our prompts.

Table 2: Number of solved tasks when using LLMs for end-to-end planning compared to a greedy best-first search within Pyperplan using the blind heuristic $h^0$, $h^{FF}$ and our LLM-generated heuristics.

| | End-to-End | | | | Pyperplan | | | | | | |
| | Gemini 2.0 | | DeepSeek | | | | Gemini 2.0 | | DeepSeek | | |
| Domain | – | Think. | V3 | R1 | $h^0$ | $h^{FF}$ | – | Think. | V3 | R1 D. | R1 |
|---|---|---|---|---|---|---|---|---|---|---|---|
| Blocksworld (90) | 2 | 40 | 2 | 17 | 6 | 24 | 35 | 37 | 45 | 34 | 66 |
| Childsnack (90) | 3 | 59 | 12 | 40 | 9 | 17 | 32 | 14 | 55 | 16 | 22 |
| Floortile (90) | 0 | 0 | 0 | 0 | 1 | 10 | 4 | 8 | 3 | 3 | 4 |
| Miconic (90) | 6 | 21 | 10 | 24 | 30 | 74 | 90 | 88 | 64 | 30 | 90 |
| Rovers (90) | 0 | 5 | 0 | 10 | 12 | 28 | 32 | 39 | 34 | 32 | 32 |
| Sokoban (90) | 0 | 14 | 0 | 8 | 24 | 31 | 31 | 32 | 31 | 24 | 30 |
| Spanner (90) | 6 | 39 | 21 | 47 | 30 | 30 | 30 | 30 | 69 | 30 | 70 |
| Transport (90) | 2 | 24 | 3 | 28 | 8 | 29 | 42 | 57 | 42 | 45 | 59 |
| Sum (720) | 19 | 202 | 48 | 174 | 120 | 243 | 296 | 305 | 343 | 214 | 373 |

**Comparisons to Domain-Independent Heuristics in Pyperplan**   We now compare the LLM-generated heuristics to two baselines: breadth-first search ($h^0$), which uses no heuristic guidance,[3] and GBFS with the $h^{FF}$ heuristic [39], which is one of the most commonly used heuristics for satisficing planning [e.g., 8, 13, 26]. These two baselines are also implemented in Pyperplan, which allows us to evaluate exactly the impact of the generated heuristics. As the right part of Table 2 shows, all LLM-generated heuristics outperform $h^0$ and $h^{FF}$ regarding total coverage, except for the distilled version of DeepSeek R1. In all but one domain (Floortile), the best performing heuristic is an LLM-generated one.

DeepSeek R1 heuristics have the highest coverage with 373 solved tasks, while DeepSeek V3 places second with 343 solved tasks. Gemini 2.0 Flash Thinking solves 68 fewer tasks (total 305). The best baseline $h^{FF}$ solves only 234 tasks. The selected DeepSeek R1 heuristic is particularly impressive in the Blocksworld domain, where it solves almost 40% more tasks as the second best heuristic (DeepSeek V3).

The non-reasoning models—Gemini 2.0 Flash and DeepSeek V3—perform worse than their reasoning counterparts. DeepSeek R1 Distill Qwen 14B heuristics are the only LLM-based models that underperform in comparison to $h^{FF}$. However, this is mostly due to its low coverage in the Miconic domain. In fact, DeepSeek R1 Distill Qwen 14B outperforms $h^{FF}$ in 3 domains, while being outperformed in 4. This shows that the smaller distilled models might be competitive with existing heuristics in some domains.

Figure 2a compares the plan lengths obtained with $h^{FF}$ and the heuristics generated by DeepSeek R1. In general, the two methods yield plans with similar lengths. The only domains where one approach has a clear edge are Blocksworld, where the DeepSeek R1 plans are consistently shorter, and Miconic, where DeepSeek R1 plans get longer than $h^{FF}$ plans as the tasks get larger.

Last, we compare the informativeness of the traditional and the LLM-generated heuristics by inspecting the number of states expanded during the search. Ideally, a heuristic only expands states traversed by a plan. Figure 2b compares expansions between $h^{FF}$ and DeepSeek R1 heuristics. The results vary by domain: DeepSeek R1 has an edge in Blocksworld, Spanner, Transport and in most of the Childsnack tasks, whereas $h^{FF}$ is more informative in Floortile, Rovers and Sokoban. In all the domains where DeepSeek R1 expands fewer states than $h^{FF}$, it also solves many more tasks than $h^{FF}$. On the flip side, the only domain where $h^{FF}$ expands fewer states and solves many more tasks than DeepSeek R1 is Floortile. In the Rovers domain, DeepSeek R1 has higher coverage than $h^{FF}$, while in Sokoban $h^{FF}$ solves only two more tasks. This indicates that despite being less informed in these two domains, the heuristics generated by DeepSeek R1 perform better because they are more efficient to compute.

---

[3]This is identical to running GBFS with the blind heuristic $h^0$, where $h^0(s) = 0$ iff $s$ is a goal state and $h^0(s) = 1$ otherwise.

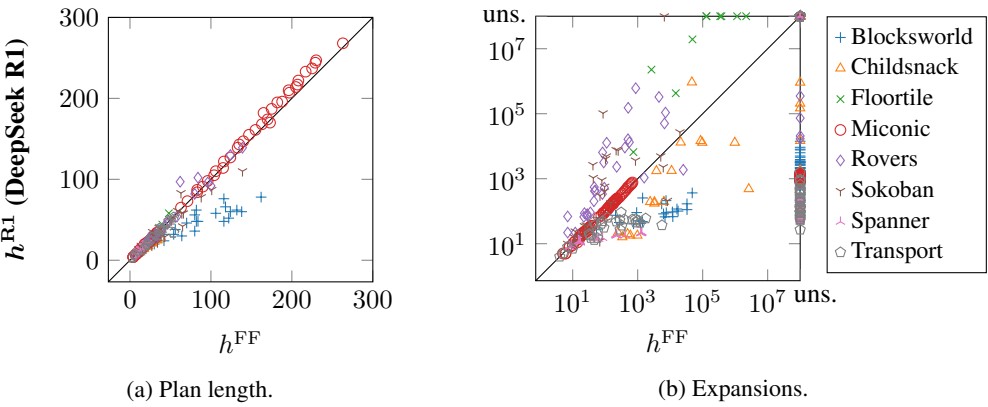

(a) Plan length.

(b) Expansions.

Figure 2: Comparison of plan length and expansions for $h^{\mathrm{FF}}$ and the heuristics generated by DeepSeek R1. We only show plan lengths up to 300 (only plans in Miconic exceed this limit).

**Comparisons to End-to-End Plan Generation with LLMs**  We also analyze the end-to-end plan generation capabilities of LLMs. We prompt the LLMs with general instructions, the PDDL files for the domain and the specific task, and a checklist of common pitfalls. To guide the model, the prompt also includes two examples: a PDDL task and PDDL domain for Gripper and Logistics (the same ones used in our prompt to generate heuristics), along with a complete optimal plan for each of the two tasks. The LLM is then asked to generate a plan, which is subsequently validated using the automated plan validation tool VAL [40]. Appendix C shows an example of our end-to-end prompt for the smallest Blocksworld task in our test set.

The left part of Table 2 shows that DeepSeek R1 solves 174 tasks and Gemini 2.0 Flash Thinking solves 202 tasks, both outperforming GBFS with the blind heuristic ($h^0$) in Pyperplan. However, both LLMs solve fewer tasks than all LLM-generated heuristics in Pyperplan, including the small DeepSeek R1 Distill Qwen 14B. With the end-to-end plan generation, the cost of using the LLMs also increases significantly. The cost for using V3 increases from \$0.25 to \$2.79, while the cost of R1 increases from \$6.12 to \$13.62. The end-to-end experiment requires 720 LLM calls (one for each of the 90 tasks across 8 domains), compared to 200 calls when generating 25 heuristics per domain. In the case of DeepSeek R1, the end-to-end experiment takes several days to complete, while the heuristics can be generated in a few hours. Furthermore, heuristics are reusable for solving new tasks, whereas end-to-end plans are not, making heuristics a more versatile and cost-effective approach.

We now compare the performance of the same LLM when used to solve two different problems: end-to-end planning versus heuristic generation. For the reasoning models Gemini 2.0 Flash Thinking and DeepSeek R1, using them for heuristic generation increases the number of solved tasks from 202 and 174 (end-to-end) to 305 and 373 (heuristic generation), respectively. The increase in total coverage is even more significant for non-reasoning models: with Gemini 2.0 Flash, the number of solved tasks jumps from 19 (end-to-end) to 296 (heuristic generation), and with DeepSeek V3 it increases from 48 (end-to-end) to 343 (heuristic generation). This shows that our heuristic generation approach can bridge the performance gap between expensive reasoning models and cheaper non-reasoning ones, while also increasing the overall number of solved tasks.

**Comparison to State-of-the-Art Heuristics Implemented in C++**  Our experimental setup has an obvious caveat: we are using Pyperplan, which is an educational, unoptimized Python planner, while all state-of-the-art planners are implemented in compiled languages such as C++. For example, the winners of all tracks of the last IPC, in 2023, are implemented in C++ [73]. Moreover, all of these planners are implemented on top of the Fast Downward planning system [35]. Even though Python is slower than C++ and uses more memory, we compare our best method, GBFS in Pyperplan using DeepSeek R1 heuristics, to GBFS in Fast Downward using the many satisficing heuristics implemented in the planner: the goal-count heuristic $h^{\mathrm{GC}}$ [23]; the landmark count heuristic $h^{\mathrm{lmc}}$ [56, 9]; the C++ implementation of the FF heuristic $h^{\mathrm{FF}}$ [39]; the context-enhanced additive heuristic $h^{\mathrm{cea}}$ [36]; the causal graph heuristic $h^{\mathrm{cg}}$ [34]; and the additive heuristic $h^{\mathrm{add}}$ [6]. We also compare it to $h^{\mathrm{WLF}}_{\mathrm{GPR}}$ [11], which uses Gaussian Process Regression [53] over features derived with the Weisfeiler-

Table 3: Coverage for different heuristics implemented in Fast Downward, and in Pyperplan. Heuristics $h^{\text{V3}}$ and $h^{\text{R1}}$ indicate the heuristics generated by DeepSeek V3 and by DeepSeek R1.

| Domain | Fast Downward (C++) | | | | | | | Pyperplan | | |
|---|---|---|---|---|---|---|---|---|---|---|
| | $h^{\text{GC}}$ | $h^{\text{lmc}}$ | $h^{\text{FF}}$ | $h^{\text{cea}}$ | $h^{\text{cg}}$ | $h^{\text{add}}$ | $h^{\text{WLF}}_{\text{GPR}}$ | $h^{\text{FF}}$ | $\mathbf{h^{V3}}$ | $\mathbf{h^{R1}}$ |
| Blocksworld (90) | 32 | 39 | 27 | 40 | 34 | 44 | 72 | 24 | 45 | 66 |
| Childsnack (90) | 23 | 13 | 25 | 29 | 29 | 29 | 31 | 17 | 55 | 22 |
| Floortile (90) | 3 | 3 | 12 | 10 | 7 | 14 | 2 | 10 | 3 | 4 |
| Miconic (90) | 90 | 90 | 90 | 79 | 90 | 90 | 90 | 74 | 64 | 90 |
| Rovers (90) | 38 | 41 | 34 | 36 | 39 | 33 | 37 | 28 | 34 | 32 |
| Sokoban (90) | 42 | 43 | 36 | 33 | 35 | 33 | 38 | 31 | 31 | 30 |
| Spanner (90) | 30 | 30 | 30 | 30 | 30 | 30 | 73 | 30 | 69 | 70 |
| Transport (90) | 36 | 36 | 41 | 49 | 54 | 51 | 28 | 29 | 42 | 59 |
| Sum (720) | 294 | 295 | 295 | 306 | 318 | 324 | 371 | 243 | 343 | 373 |

Leman algorithm [69] to learn domain-dependent heuristics, and is considered the state-of-the-art in classical planning for heuristic learning. $h^{\text{WLF}}_{\text{GPR}}$ is also implemented on top of Fast Downward.

From now on, we denote the heuristics generated by DeepSeek V3 as $h^{\text{V3}}$ and the ones by DeepSeek R1 as $h^{\text{R1}}$. Table 3 shows that GBFS in Pyperplan with $h^{\text{V3}}$ and with $h^{\text{R1}}$ solves more tasks in total than *any of the traditional Fast Downward heuristics*. Moreover, $h^{\text{R1}}$ is also *competitive with the state-of-the-art,* $h^{\text{WLF}}_{\text{GPR}}$, and achieves slightly higher total coverage. This is quite an unexpected result, as Pyperplan is not as engineered and receives little attention compared to Fast Downward. It indicates that the heuristics generated by DeepSeek R1 are indeed powerful, being capable of surpassing the performance gap between Python and C++ implementations. Additionally, it shows that even the non-reasoning model DeepSeek V3 can outperform classical planners with our method.

**Ablation Study**  We now analyze the impact of the different components in our prompt in an ablation study. For this, we use Gemini 2.0 Flash Thinking to generate 25 heuristics for several variants of our prompt, each of which alters or removes a single component of the original prompt. To reduce the effects of randomness, we run *all* generated heuristics on the test set, instead of selecting only the single best heuristic. We limit each GBFS run in this experiment to 5 minutes.

Table 4 shows the results. For the first ablation (second column in the table), we replace the long instructions at the beginning of the prompt by a simple request to generate a heuristic function, omitting details such as the request to minimize expansions. For the "Heuristics" ablation, we replace the provided domain-dependent example heuristics with domain-independent heuristics available in Pyperplan, namely the goal-count heuristic and four heuristics based on delete-relaxation [1].[4] For all other ablations, we remove the corresponding component from the prompt.

Comparing the heuristics with the best coverage score among all ablations, we see that the original prompt (solving 423 tasks) outperforms all other ablations. This score is $38.7\%$ higher than the one obtained by using our method of selecting heuristics based on performance on the training set (see Table 2), and it even surpasses $h^{\text{R1}}$. This shows that while our heuristic selection method is good enough to outperform state-of-the-art planners, it is still not the optimal selection strategy. We leave the challenge of devising better selection strategies for future work.

Focusing on the worst possible coverage and the average coverage, our original prompt performs worse than other ablations. This result is to be expected: as our method aims to have a very diverse pool by generating multiple heuristics at a high temperature, we expect a great variance in performance. This is supported by the large standard deviation in our results. To obtain more consistent results, one could simply use a lower temperature, for example. However, this is not desirable: it is much better to have a single well-performing heuristic than several average ones.

---

[4]We do so because completely removing the example heuristics yields a coverage of 0 for all generated heuristics in all domains.

Table 4: Ablation study on the impact of different prompt components using Gemini 2.0 Flash Thinking. Components are ordered according to the original prompt. The symbol "−" indicates that the component was removed, and the symbol "⇆" indicates that the component was replaced with a weaker version (see text for details). A failed heuristic is a heuristic that solves no tasks.

| | Original Prompt | ⇆ Instruction | − PDDL Domain | − PDDL Tasks | ⇆ Heuristics | − State Repr. | − Static Repr. | − Pyperplan Code | − Checklist |
|---|---|---|---|---|---|---|---|---|---|
| Best Coverage | 423 | 359 | 368 | 401 | 404 | 402 | 404 | 401 | 382 |
| Worst Coverage | 114 | 138 | 118 | 90 | 126 | 133 | 92 | 110 | 63 |
| Avg. Coverage | 267.0 | 242.5 | 237.3 | 261.5 | 263.2 | 253.7 | 243.8 | 270.0 | 260.0 |
| Std. Deviation | ±94.3 | ±59.6 | ±70.4 | ±86.8 | ±87.3 | ±85.0 | ±95.0 | ±97.4 | ±87.8 |
| Failed Heuristics | 64 | 57 | 52 | 42 | 64 | 68 | 57 | 97 | 57 |

For average coverage, we exclude heuristics that fail by crashing or not solving any task in our analysis. In this analysis, most of the components of our prompt are beneficial, with the most impactful component being the PDDL domain description. The only component whose removal actually slightly increases the average coverage is the Pyperplan code. However, its removal also increases the number of heuristics (out of 200) that fail by almost $50\%$. These results show that all prompt components are important for generating high-quality heuristics.

**Examples of LLM-Generated Heuristic Functions**   We illustrate the heuristic functions generated by DeepSeek R1 using Blocksworld and Spanner (see Appendix B). Both domains have polynomial algorithms [30]. In Blocksworld, $n$ blocks are rearranged by moving blocks from stacks. The selected heuristic identifies misplaced goal blocks $A$, adding 1 to the heuristic value plus 2 for each block $B$ on top of $A$, using an auxiliary function for stack traversal. In Spanner, an agent traverses a one-way corridor to pick spanners (each used once) and tighten $n$ nuts; moving without a required spanner can lead to an unsolvable state. The Spanner heuristic greedily assigns the closest available spanner to each loose nut. The cost calculation depends on whether the spanner is already picked up (agent-to-nut distance + 1) or not (agent-to-spanner + spanner-to-nut distances + 2), with a large penalty for unassigned nuts. This heuristic precomputes all-pairs shortest paths using breadth-first search during initialization. We describe and analyze the generated heuristics for the remaining domains in Appendix B.

**Memorization vs. Reasoning**   To verify whether our method is robust to new domains, we run three additional experiments testing if the LLMs retrieve memorized heuristics or reason over the provided PDDL domain. Each experiment introduces a new PDDL domain.

We begin with a *qualitative* experiment: a variant of the Spanner domain. In the original Spanner domain, an agent traverses a one-way corridor toward a gate while picking up spanners. It must tighten $n$ nuts to open the gate, and spanners break after a single use. Consequently, the agent must collect at least $n$ spanners. In this variant, spanners no longer break after use. If the LLM merely recalled a standard Spanner heuristic, it would likely ignore this change and produce longer plans. Instead, the generated heuristic adapts to the new semantics. The best LLM-generated heuristic is perfect for this modified domain, yielding optimal plan lengths for all states. This suggests the LLM reasoned about the given domain rather than retrieving a memorized solution.

Next, we *obfuscate* Blocksworld following Valmeekam et al. [78]: all symbols (action names, predicate symbols, objects) are renamed to random strings. We refer to this as Obfuscated Blocksworld. In this experiment, $h^{\mathrm{FF}}$ (in Fast Downward) solves a similar number of tasks in both versions, 27 for Blocksworld and 28 for Obfuscated Blocksworld. This is expected, as $h^{\mathrm{FF}}$ does not rely on the semantics of the PDDL names. In contrast, $h^{\mathrm{R1}}$ (in Pyperplan) solves 66 tasks in Blocksworld but only 40 in Obfuscated Blocksworld. This indicates that while semantic cues from the PDDL names are useful, the LLM can still reason about the domain's logical structure without token semantics. We note that this is a highly adversarial setting for LLMs, which are trained to exploit token semantics rather than operate on random identifiers.

Finally, we use a completely new domain, *Rod-Rings*, which has not been released online nor exposed to an LLM. The domain features sticks with stacks of rings and a single "held" ring. Two moves are allowed: (a) place the held ring at the bottom of a stick (swapping with the current top ring), or (b) place the held ring at the top of a stick (swapping with the bottom ring). The goal is to arrange specified rings on specified sticks in a defined order. To avoid leakage, we use an OpenAI model (o3) via their API for this experiment. For Rod-Rings, $h^{o3}$ guiding a GBFS in Pyperplan solves 58 tasks, nearly matching Fast Downward with $h^{FF}$ (59 tasks).[5] This suggests the LLM reasons about this unseen domain.

Together, these three experiments indicate that the success of our method is due to the LLMs' ability to generate domain-dependent heuristic functions by reasoning about the logical structure of the domain.

## 5   Related Work

The combination of planning and learning to create heuristic functions has a long tradition [61, 12, 60, 2]. There are two main paradigms for learning heuristic functions in classical planning: task-dependent [20, 21, 49, 4] and domain-dependent [68, 71, 10, 31]. In this paper, we consider the second paradigm. Currently, the strongest approach in domain-dependent heuristic learning is $h_{GPR}^{WLF}$ [11], which we compare to above.

Recently, LLMs entered the picture. Yet, Valmeekam et al. [78] show that LLMs cannot reliably solve even small classical planning tasks when used for end-to-end plan generation. Moreover, techniques such as supervised fine-tuning and chain-of-thought [43] fail to generalize to out-of-distribution tasks [5, 72]. This holds even when using more advanced prompting techniques such as Tree of Thoughts [81] or Algorithm of Thoughts (AoT) [66]. While the strongest variant of this line of work, AoT+ [67], finds valid plans for up to 82% of the tasks in their Blocksworld benchmark set, the largest task in their set only has 5 blocks [78]. In contrast, our $h^{R1}$ heuristic solves tasks with up to 216 blocks and never yields incorrect plans. As seen in our experiments, even LLMs explicitly designed for reasoning tasks are not competitive with state-of-the-art planners.

Nonetheless, Rossetti et al. [59] and Huang et al. [41] show that LLMs trained to plan can achieve competitive performance when training and test sets share the same distribution. Furthermore, LLMs can also help to solve classical planning tasks when combined with other techniques. For example, there is an extensive body of work exploring the potential of LLMs to convert problems described in natural language into PDDL tasks [e.g., 29, 24, 48, 46].

The most closely related approaches to ours are those that use LLMs to generate code for solving planning tasks. Katz et al. [42] highlight the high computational cost of using LLMs for end-to-end plan generation, particularly when multiple inferences are required. They propose to use LLMs to generate Python code for successor generation and goal testing, which are then used with standard search algorithms. While more efficient than end-to-end LLM planning, their method requires human feedback for incorrect code and is limited to small tasks due to its reliance on uninformed search. Our approach could address this scalability issue by providing a heuristic for an informed search algorithm.

Silver et al. [70] also use LLMs to generate Python code for solving classical planning tasks. However, they focus on *generalized planning*, where the aim is to find a strategy that efficiently solves any task of a given domain. In their approach, an LLM generates a "simple" Python program that does not rely on search. The key difference to our approach is that they address a different problem: there are many planning domains, such as the Sokoban domain we use above, for which no simple strategy exists to efficiently produce plans. For such domains, heuristic functions can be useful.

The work by Tuisov et al. [77] is the most similar to ours, and we draw inspiration from their work. They also use LLMs to generate heuristic function code for automated planning, though, their focus is on numeric planning. Their approach differs from ours in three main ways. First, they require a manual translation of each PDDL domain into Rust, including the implementation of a successor generator and a goal test. The cost of translating domains to Rust prevents easy comparisons on all IPC domains. In contrast, our approach generates heuristics directly from the PDDL description with the resulting code integrated into an off-the-shelf planner. Second, their heuristics are task-dependent,

---

[5]On the eight IPC 2023 domains, $h^{o3}$ heuristics solve 354 tasks, 19 fewer than $h^{R1}$ (373 tasks).

requiring new LLM inferences for each task, while ours are domain-dependent and reusable, reducing costs. Finally, while their approach outperforms all domain-independent heuristics they compare against, their LLM-generated heuristics result in fewer expansions for only one task. This suggests that while their heuristics may be computationally faster, they are not necessarily more informative. In our experiments, the LLM-generated heuristics are often more informative than the traditional domain-independent ones.

Beyond PDDL planning, Romera-Paredes et al. [58] use a search in function space to solve combinatorial problems. Their algorithm, FunSearch, samples different initial programs, similar to our set of candidate heuristics. However, FunSearch feeds the best initial candidates back into the LLM to improve them. In contrast, our pipeline never feeds the functions back into the LLM. Although our results are already positive, this feedback loop could further strengthen our results.

## 6 Limitations

Our approach, while effective, has limitations. One such limitation is its dependency on a formal PDDL description. This dependence makes our method less general than end-to-end LLM planning approaches that can use natural language. However, recent approaches translate natural language into PDDL representations [e.g., 24, 74], and can bridge this gap.

A second limitation is that our method, which selects the heuristic, is itself heuristic. This selection does not guarantee that the chosen heuristic generalizes to the out-of-distribution test set. However, our empirical results indicate that the selected heuristic consistently outperforms other approaches, indicating its robustness.

The effectiveness of our methods also depends on a multi-component prompt. Our ablation study confirmed that while the method is robust, the best performance indeed requires all components of the prompt.

Finally, our current implementation is a proof-of-concept that uses Pyperplan, an educational planner which is significantly slower and less memory-efficient than state-of-the-art C++ planners like Fast Downward. This implementation difference means that a C++ implementation would be necessary to conduct a direct performance comparison and verify the full potential of the LLM-generated heuristics. For example, Appendix E shows that Fast Downward expands up to 669 times more states per second than Pyperplan in certain domains.

## 7 Conclusions

In this paper, we show how to use LLMs to generate domain-dependent heuristic functions for classical planning domains. Our approach uses LLMs to produce a pool of candidate heuristics, which we then evaluate on a training set in order to choose the best heuristic from the pool. The selected heuristic is then used to solve unseen out-of-distribution test tasks.

We provide a proof-of-concept implementation of this pipeline in Pyperplan, an educational classical planner written in Python. We show that our LLM-generated heuristics outperform directly using LLMs end-to-end prompted to produce plans. This difference is particularly noticeable when using non-reasoning LLMs. Comparing the Python-based heuristics, we see that our LLM-generated heuristics outperform state-of-the-art domain-independent heuristics in most of the domains of our benchmark set. In particular, heuristics generated by reasoning LLMs such as DeepSeek R1 show a significant improvement compared to the domain-independent heuristic.

We show that Pyperplan equipped with the heuristics from DeepSeek V3 ($h^{V3}$) and DeepSeek R1 ($h^{R1}$) outperform heuristics implemented in Fast Downward [35], a state-of-the-art planner written in C++. Moreover, $h^{R1}$ is also competitive with $h^{WLF}_{GPR}$ [11], the state-of-the-art in heuristic learning for classical planning implemented on top of Fast Downward. These results are surprising, as Pyperplan is much less optimized than Fast Downward, and DeepSeek R1 is not trained for a specific domain, while $h^{WLF}_{GPR}$ is. Our results show the potential of LLM-generated heuristics in classical planning as an efficient and effective approach to improve the planning capabilities of LLMs.

## Acknowledgments

We thank Elliot Gestrin for designing the Rod-Rings domain, and Malte Helmert for giving us access to the cluster of the AI group at the University of Basel.

Jendrik Seipp was supported by the Wallenberg AI, Autonomous Systems and Software Program (WASP) funded by the Knut and Alice Wallenberg Foundation. André G. Pereira acknowledges support from FAPERGS with project 21/2551-0000741-9. This study was financed in part by the *Coordenação de Aperfeiçoamento de Pessoal de Nível Superior – Brasil* (CAPES) – Finance Code 001.

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

## A    Heuristic Generation Prompt: Blocksworld

Below we show our prompt used for heuristic generation in the Blocksworld domain. The only parts of the prompt that change between domains are the names of the domain and heuristic, and the `domain-file`, `instance-file-example-1` and `instance-file-example-2` sections. To reduce the number of pages, we do not display the entire domain and instance files, but just the beginning of each. The complete domain and instance files can be found online [14].

```
1   <problem-description>
2   You are a highly-skilled professor in AI planning and a proficient Python
    ↪   programmer creating a domain-dependent heuristic function for the PDDL domain
    ↪   <domain>blocksworld</domain>. The heuristic function you create will be used
    ↪   to guide a greedy best-first search to solve instances from this domain.
    ↪   Therefore, the heuristic does not need to be admissible. For a given state,
    ↪   the heuristic function should estimate the required number of actions to reach
    ↪   a goal state as accurately as possible, while remaining efficiently computable.
    ↪   The name of the heuristic should be blocksworld1Heuristic. The heuristic
    ↪   should be efficiently computable, and it should minimize the number of
    ↪   expanded nodes during the search. Next, you will receive a sequence of file
    ↪   contents to help you with your task and to show you the definition of the
    ↪   blocks world domain.
3   </problem-description>
4
5   This is the PDDL domain file of the blocksworld domain, for which you need to
    ↪   create a domain-dependent heuristic:
6   <domain-file>
7   (define (domain blocksworld)
8   [...]
9   </domain-file>
10
11  This is an example of a PDDL instance file of the blocksworld domain:
12  <instance-file-example-1>
13  (define (problem blocksworld-01)
14  [...]
15  </instance-file-example-1>
16
17  This is a second example of a PDDL instance file of the blocksworld domain:
18  <instance-file-example-2>
19
20  (define (problem blocksworld-99)
21   (:domain blocksworld)
22   [...]
23  </instance-file-example-2>
24
25  This is the PDDL domain file of another domain, called Gripper, which serves as an
    ↪   example:
26  <gripper-domain-file>
27  (define (domain gripper-strips)
28  [...]
29  </gripper-domain-file>
30
31  This is an example of an instance file from the Gripper domain:
32  <gripper-instance-file-example>
33  (define (problem strips-gripper-x-20)
34  [...]
35  </gripper-instance-file-example>
36
37  This is an example of a domain-dependent heuristic for Gripper:
38  <code-file-heuristic-1>
39  from fnmatch import fnmatch
40  from heuristics.heuristic_base import Heuristic
41
42  class GripperHeuristic(Heuristic):
43      """
```

```python
44          A domain-dependent heuristic for the Gripper domain.
45
46          # Summary
47          This heuristic estimates the number of actions needed to transport all balls
48          from `rooma` to `roomb`.
49
50          # Assumptions:
51          - The robot has two grippers, allowing it to carry up to two balls per trip.
52          - The robot must return to rooma after each trip, except for the final trip.
53          - If the robot starts in roomb, it must move to rooma first.
54
55          # Heuristic Initialization
56          - Implicitly assume that all balls must be in `roomb` at the end.
57
58          # Step-By-Step Thinking for Computing Heuristic
59          1. Identify the number of balls still in `rooma` that need to be transported.
60          2. Determine if the robot is currently carrying balls (it may start with 1 or
            ↪    2 already).
61          3. Check whether the robot is in `rooma` or `roomb`:
62             - If in room B, it may need to drop the carried balls first before moving
                ↪    to A.
63             - If in room A, it can immediately begin planning the transport.
64          4. Handle the case where the robot starts with balls in the grippers:
65             - If carrying 2 balls, it should move to B, drop them, and return to A.
66             - If carrying 1 ball and an odd number remains in `rooma`, it may pick up
                ↪    another ball before moving.
67             - If carrying 1 ball and an even number remains, it transports the single
                ↪    ball first.
68          5. Compute the number of full two-ball trips needed:
69             - This is `balls_in_rooma // 2` (since up to 2 balls are moved per trip).
70             - Each full two-ball trip costs 6 actions (except for the last trip).
71          6. Handle the last remaining ball (if the total number of balls is odd):
72             - If one ball is left, the robot moves to A, picks it up, moves to B, and
                ↪    drops it.
73          """
74
75      def __init__(self, task):
76          """Initialize the heuristic by extracting goal conditions and static
            ↪    facts."""
77          # The set of facts that must hold in goal states. We assume that all balls
            ↪    must be in `roomb` at the end.
78          self.goals = task.goals
79          # Static facts are not needed for this heuristic.
80          static_facts = task.static
81
82      def __call__(self, node):
83          """Estimate the minimum cost to transport all remaining balls from room A
            ↪    to room B."""
84          state = node.state
85
86          def match(fact, *args):
87              """
88              Utility function to check if a PDDL fact matches a given pattern.
89              - `fact`: The fact as a string (e.g., "(at ball1 rooma)").
90              - `args`: The pattern to match (e.g., "at", "*", "rooma").
91              - Returns `True` if the fact matches the pattern, `False` otherwise.
92              """
93              parts = fact[1:-1].split()  # Remove parentheses and split into
                ↪    individual elements.
94              return all(fnmatch(part, arg) for part, arg in zip(parts, args))
95
96          # Count how many balls are currently in room A.
97          balls_in_room_a = sum(1 for fact in state if match(fact, "at", "*",
            ↪    "rooma"))
98
```

```
 99              # Count the number of balls currently held by the robot.
100              balls_in_grippers = sum(1 for fact in state if match(fact, "carry", "*",
      ↪          "*"))
101
102              # Check if the robot is in room A.
103              robot_in_room_a = "(at-robby rooma)" in state
104
105              # Define the cost of each individual action for readability.
106              move_cost = 1  # Moving between rooms.
107              pick_cost = 1  # Picking up a ball.
108              drop_cost = 1  # Dropping a ball.
109
110              total_cost = 0  # Initialize the heuristic cost.
111
112              # Handle cases where the robot is already carrying balls.
113              if robot_in_room_a:
114                  if balls_in_grippers == 2:
115                      # Both grippers are full, so move to room B and drop both balls.
116                      total_cost += move_cost + 2 * drop_cost
117                  elif balls_in_grippers == 1 and balls_in_room_a % 2 == 1:
118                      # Pick one more ball to fill both grippers, then move and drop
      ↪                  both.
119                      total_cost += pick_cost + move_cost + 2 * drop_cost
120                      balls_in_room_a -= 1  # Since we moved one extra ball.
121                  elif balls_in_grippers == 1 and balls_in_room_a % 2 == 0:
122                      # Move with one ball and drop it, leaving an even number of balls.
123                      total_cost += move_cost + drop_cost
124              else:
125                  # If the robot is in room B, it must drop any carried balls.
126                  total_cost += balls_in_grippers * drop_cost
127
128              if balls_in_room_a > 0:
129                  # Move back to room A to continue transporting balls.
130                  total_cost += move_cost
131
132                  # Compute the number of trips with two balls.
133                  num_two_ball_trips = balls_in_room_a // 2
134
135                  # Each trip includes: 2 picks, 1 move to B, 2 drops and 1 move back to
      ↪              A (except for the last trip).
136                  total_cost += num_two_ball_trips * (2 * pick_cost + move_cost + 2 *
      ↪              drop_cost + move_cost) - 1
137
138                  # If there's a single ball left after the two-ball trips, go back to A
      ↪              and move the ball by itself.
139                  if balls_in_room_a % 2 == 1:
140                      total_cost += move_cost + pick_cost + move_cost + drop_cost
141
142          # Return the estimated cost to goal state.
143          return total_cost
144  </code-file-heuristic-1>
145
146  This is the PDDL domain file of another domain, called Logistics, to serve as a
      ↪  second example:
147  <logistics-domain-file>
148  (define (domain logistics-strips)
149    (:requirements :strips)
150    (:predicates          (OBJ ?obj)
151                          (TRUCK ?truck)
152                          (LOCATION ?loc)
153                      (AIRPLANE ?airplane)
154                      (CITY ?city)
155                      (AIRPORT ?airport)
156                      (at ?obj ?loc)
157                      (in ?obj1 ?obj2)
```

```
158                        (in-city ?obj ?city))
159
160    (:action LOAD-TRUCK
161      :parameters
162       (?obj
163        ?truck
164        ?loc)
165      :precondition
166       (and (OBJ ?obj) (TRUCK ?truck) (LOCATION ?loc)
167       (at ?truck ?loc) (at ?obj ?loc))
168      :effect
169       (and (not (at ?obj ?loc)) (in ?obj ?truck)))
170    [...]
171    </logistics-domain-file>
172
173    This is an example of an instance file from the Logistics domain:
174    <logistics-instance-file-example>
175    (define (problem strips-log-y-5)
176        (:domain logistics-strips)
177        (:objects package5 package4 package3 package2 package1 city8
178                  city7 city6 city5 city4 city3 city2 city1 truck15 truck14
179                  truck13 truck12 truck11 truck10 truck9 truck8 truck7 truck6
180                  truck5 truck4 truck3 truck2 truck1 plane1 city8-2 city8-1
181                  city7-2 city7-1 city6-2 city6-1 city5-2 city5-1 city4-2
182                  city4-1 city3-2 city3-1 city2-2 city2-1 city1-2 city1-1
183                  city8-3 city7-3 city6-3 city5-3 city4-3 city3-3 city2-3
184                  city1-3)
185        (:init (obj package5)
186               (obj package4)
187    [...]
188    </logistics-instance-file-example>
189
190    This is an example of a domain-dependent heuristic for Logistics:
191    <code-file-heuristic-2>
192    from fnmatch import fnmatch
193    from heuristics.heuristic_base import Heuristic
194
195
196    def get_parts(fact):
197        """"Extract the components of a PDDL fact by removing parentheses and splitting
           ↪   the string."""
198        return fact[1:-1].split()
199
200
201    def match(fact, *args):
202        """
203        Check if a PDDL fact matches a given pattern.
204
205        - `fact`: The complete fact as a string, e.g., "(in-city airport1 city1)".
206        - `args`: The expected pattern (wildcards `*` allowed).
207        - Returns `True` if the fact matches the pattern, else `False`.
208        """
209        parts = get_parts(fact)
210        return all(fnmatch(part, arg) for part, arg in zip(parts, args))
211
212
213    class LogisticsHeuristic(Heuristic):
214        """
215        A domain-dependent heuristic for the Logistics domain.
216
217        # Summary
218        The heuristic estimates the number of necessary actions (load, unload, and
           ↪   transport) in order to transport each package to its goal based on its
           ↪   current state.
219
```

```
220        # Assumptions
221        - Packages can be on the ground, in a truck, or in a plane.
222        - Trucks move within a city, while planes move between cities.
223        - If a package is already at the goal, no extra actions are needed.
224
225        # Heuristic Initialization
226        - Extract the goal locations for each package and the static facts (e.g.,
           ↪  `in-city` relationships and airport locations) from the task.
227
228        # Step-by-Step Thinking for Computing the Heuristic Value
229        Below is the thought process for computing the heuristic for a given state:
230
231        1. Extract Relevant Information:
232        - Identify the current location of every package.
233        - Identify whether a package is inside a vehicle (truck or plane), and if so,
           ↪  find the physical location of that vehicle.
234
235        2. Distinguish Between Intra-city and Inter-city Transport:
236        - Determine the current city and goal city for each package by checking its
           ↪  location-to-city mapping.
237        - If the current city is the same as the goal city, follow the intra-city
           ↪  package movement rules.
238        - If the current city is different from the goal city, follow the inter-city
           ↪  package movement rules.
239
240        3. Handle Intra-city Transport:
241        - If the package is already at its goal location, no action is required.
242        - If the package is in a plane, it must be unloaded.
243        - If the package is not in a truck and not already at its goal, it must be
           ↪  loaded into a truck.
244        - If the package is in a truck or not yet at its final location, it must be
           ↪  unloaded from the truck at the goal.
245
246        4. Handle Inter-city Transport:
247        - Step 1: Move the package to the airport in the current city.
248            - If the package is not inside a truck and not at an airport, it must be
               ↪  loaded into a truck.
249            - If the package is not at an airport or inside a truck, it must be
               ↪  unloaded from the truck at the airport.
250        - Step 2: Fly the package to the destination city.
251            - If the package is not in a plane, it must be loaded into a plane.
252            - The package must always be unloaded from the plane at the airport of the
               ↪  destination city.
253        - Step 3: Move the package from the airport to its final location.
254            - If the goal location is not an airport, the package must be loaded into
               ↪  a truck at the airport.
255            - Finally, the package must be unloaded from the truck at the goal
               ↪  location.
256
257        5. Summing the Actions:
258        - The total heuristic value is the sum of all necessary actions.
259        - Loading and unloading costs are counted exactly based on the required
           ↪  actions.
260        - Transport movements (trucks or planes) are counted only when necessary.
261        """
262
263        def __init__(self, task):
264            """
265            Initialize the heuristic by extracting:
266            - Goal locations for each package.
267            - Static facts (`in-city` relationships and airport locations).
268            """
269            self.goals = task.goals  # Goal conditions.
270            static_facts = task.static  # Facts that are not affected by actions.
271
```

```
272          # Map locations to their respective cities using "in-city" relationships.
273          self.location_to_city = {
274              get_parts(fact)[1]: get_parts(fact)[2]
275              for fact in static_facts
276              if match(fact, "in-city", "*", "*")
277          }
278
279          # Identify all airport locations.
280          self.airports = {
281              get_parts(fact)[1]
282              for fact in static_facts
283              if match(fact, "airport", "*")
284          }
285
286          # Store goal locations for each package.
287          self.goal_locations = {}
288          for goal in self.goals:
289              predicate, *args = get_parts(goal)
290              if predicate == "at":
291                  package, location = args
292                  self.goal_locations[package] = location
293
294      def __call__(self, node):
295          """Compute an estimate of the minimal number of required actions."""
296          state = node.state  # Current world state.
297
298          # Track where packages and vehicles are currently located.
299          current_locations = {}
300          for fact in state:
301              predicate, *args = get_parts(fact)
302              if predicate in ["at", "in"]:  # Track both direct location and inside
       ↪   vehicle.
303                  obj, location = args
304                  current_locations[obj] = location
305
306          total_cost = 0  # Initialize action cost counter.
307
308          for package, goal_location in self.goal_locations.items():
309              # Get the current location of the package (could be a city location,
       ↪   truck or plane).
310              current_location = current_locations[package]
311
312              # Check if the package is inside a vehicle.
313              in_vehicle = current_location not in self.location_to_city
314
315              if in_vehicle:
316                  # Identify type of vehicle (truck or plane).
317                  in_plane = current_location.startswith("plane")
318                  in_truck = current_location.startswith("truck")
319                  assert in_plane ^ in_truck, f"Invalid state: {current_location}"
320
321                  # Retrieve the physical location of the vehicle.
322                  current_location = current_locations[current_location]
323              else:
324                  in_plane = False
325                  in_truck = False
326
327              # Get the city of the package's current location and goal.
328              current_city = self.location_to_city[current_location]
329              goal_city = self.location_to_city[goal_location]
330
331              # Intra-city Transport (Same City)
332              if current_city == goal_city:
333                  if in_plane:
334                      total_cost += 1  # Unload from the plane.
```

```
335
336                         if current_location != goal_location and not in_truck:
337                             total_cost += 1  # Load into a truck.
338
339                         if current_location != goal_location or in_truck:
340                             total_cost += 1  # Unload from the truck.
341
342                     # Inter-city Transport (Different Cities)
343                     else:
344                         # Step 1: Move to the airport in the current city.
345                         if current_location not in self.airports and not in_truck:
346                             total_cost += 1  # Load into a truck.
347
348                         if current_location not in self.airports or in_truck:
349                             total_cost += 1  # Unload from the truck at the airport.
350
351                         # Step 2: Fly to the destination city.
352                         if not in_plane:
353                             total_cost += 1  # Load into a plane.
354
355                         total_cost += 1  # Unload from the plane.
356
357                         # Step 3: Transport from airport to the goal (if required).
358                         if goal_location not in self.airports:
359                             total_cost += 1  # Load into a truck at destination airport.
360                             total_cost += 1  # Unload at the destination.
361
362             return total_cost
363 </code-file-heuristic-2>
364
365 This is how an example state from the blocksworld domain is represented internally
    ↪ by the planner. Note that PDDL facts are represented as strings, for example
    ↪  '(predicate_name object1 object2)'.
366 <state>
367 frozenset({'(on-table b9)', '(on b8 b9)', '(on-table b2)', '(on b10 b2)', '(on b11
    ↪ b10)', '(on b4 b8)', '(on b6 b11)', '(on b5 b6)', '(on b14 b5)', '(on b13 b4)',
    ↪ '(on b12 b13)', '(on b1 b12)', '(on b7 b1)', '(clear b3)', '(on b3 b7)',
    ↪ '(holding b15)', '(clear b14)'})
368 </state>
369
370 This is an example for how the static information is represented internally by the
    ↪  planner:
371 <static>
372 frozenset()
373 </static>
374
375 This is the source code for representing operators and tasks in the planner:
376 <code-file-task>
377 """
378 Classes for representing a STRIPS planning task
379 """
380
381
382 class Operator:
383     """
384     The preconditions represent the facts that have to be true
385     before the operator can be applied.
386     add_effects are the facts that the operator makes true.
387     delete_effects are the facts that the operator makes false.
388     """
389
390     def __init__(self, name, preconditions, add_effects, del_effects):
391         self.name = name
392         self.preconditions = frozenset(preconditions)
393         self.add_effects = frozenset(add_effects)
```

```
394            self.del_effects = frozenset(del_effects)

395
396     def applicable(self, state):
397         """
398         Operators are applicable when their set of preconditions is a subset
399         of the facts that are true in "state".
400
401         @return True if the operator's preconditions is a subset of the state,
402                 False otherwise
403         """
404         return self.preconditions <= state

405
406     def apply(self, state):
407         """
408         Applying an operator means removing the facts that are made false
409         by the operator from the set of true facts in state and adding
410         the facts made true.
411
412         Note that therefore it is possible to have operands that make a
413         fact both false and true. This results in the fact being true
414         at the end.
415
416         @param state The state that the operator should be applied to
417         @return A new state (set of facts) after the application of the
418                 operator
419         """
420         assert self.applicable(state)
421         assert type(state) in (frozenset, set)
422         return (state - self.del_effects) | self.add_effects

423
424     def __eq__(self, other):
425         return (
426             self.name == other.name
427             and self.preconditions == other.preconditions
428             and self.add_effects == other.add_effects
429             and self.del_effects == other.del_effects
430         )

431
432     def __hash__(self):
433         return hash((self.name, self.preconditions, self.add_effects,
            ↪  self.del_effects))

434
435     def __str__(self):
436         s = "%s\n" % self.name
437         for group, facts in [
438             ("PRE", self.preconditions),
439             ("ADD", self.add_effects),
440             ("DEL", self.del_effects),
441         ]:
442             for fact in facts:
443                 s += "  {}: {}\n".format(group, fact)
444         return s

445
446     def __repr__(self):
447         return "<Op %s>" % self.name

448
449
450  class Task:
451     """
452     A STRIPS planning task
453     """

454
455     def __init__(self, name, facts, initial_state, goals, operators, static):
456         """
457         @param name The task's name
```

```
458        @param facts A set of all the fact names that are valid in the domain
459        @param initial_state A set of fact names that are true at the beginning
460        @param goals A set of fact names that must be true to solve the problem
461        @param operators A set of operator instances for the domain
462        @param static_info A set of facts that are true in every state
463        """
464        self.name = name
465        self.facts = facts
466        self.initial_state = initial_state
467        self.goals = goals
468        self.operators = operators
469        self.static = static
470
471    def goal_reached(self, state):
472        """
473        The goal has been reached if all facts that are true in "goals"
474        are true in "state".
475
476        @return True if all the goals are reached, False otherwise
477        """
478        return self.goals <= state
479
480    def get_successor_states(self, state):
481        """
482        @return A list with (op, new_state) pairs where "op" is the applicable
483        operator and "new_state" the state that results when "op" is applied
484        in state "state".
485        """
486        return [(op, op.apply(state)) for op in self.operators if
                ↪  op.applicable(state)]
487
488    def __str__(self):
489        s = "Task {0}\n  Vars:  {1}\n  Init:  {2}\n  Goals: {3}\n  Ops:   {4}"
490        return s.format(
491            self.name,
492            ", ".join(self.facts),
493            self.initial_state,
494            self.goals,
495            "\n".join(map(repr, self.operators)),
496        )
497
498    def __repr__(self):
499        string = "<Task {0}, vars: {1}, operators: {2}>"
500        return string.format(self.name, len(self.facts), len(self.operators))
501 </code-file-task>
502
503 Provide only the Python code of the domain-dependent heuristic for the blocksworld
    ↪  domain. Here is a checklist to help you with your code:
504 1) The code for extracting objects from facts remembers to ignore the surrounding
    ↪  brackets.
505 2) The heuristic is 0 only for goal states.
506 3) The heuristic value is finite for solvable states.
507 4) All used modules are imported.
508 5) The information from static facts is extracted into suitable data structures in
    ↪  the constructor.
509 6) Provide a detailed docstring explaining the heuristic calculation. For this,
    ↪  divide the docstring into sections "Summary", "Assumptions", "Heuristic
    ↪  Initialization" and "Step-By-Step Thinking for Computing Heuristic".
```

## B   Generated Heuristics (Selection)

We present the selected DeepSeek R1 heuristics from our experiments. We provide descriptions and source code for the Blocksworld and Spanner domains, and only the descriptions for the rest.

## B.1 DeepSeek R1 Heuristic for Blocksworld

In the *Blocksworld* domain, stacks of $n$ blocks must be rearranged from an initial state to a goal condition. The available actions move blocks that are on top of a stack onto a different stack or the table. Blocksworld is *2-approximable* [30]: simply "destroy" all stacks (by placing their blocks onto the table) and then build the goal stacks.

The best heuristic generated by the LLM does not use this method though. It first computes for each block $A$ mentioned in the goal whether $A$ is misplaced and, if so, adds 2 to the heuristic value for each block $B$ on top of $A$, plus 1. For this, the heuristic uses an auxiliary function that traverses the stack on top of $A$. It is easy to see that this heuristic can overestimate the optimal plan length.

```python
from fnmatch import fnmatch
from collections import defaultdict
from heuristics.heuristic_base import import Heuristic

class blocksworld9Heuristic(Heuristic):
    """
    A domain-dependent heuristic for the Blocksworld domain.

    # Summary
    This heuristic estimates the number of actions required to achieve the goal by
    considering the number of blocks that need to be moved and the blocks above
    them in the current state. For each block not in its goal position, the cost
    is 2 times the number of blocks above it plus 2. Held blocks not in the goal
    position add 1 to the cost.

    # Assumptions
    - The goal specifies the required 'on' and 'on-table' predicates for certain
      blocks.
    - Blocks not mentioned in the goal do not affect the heuristic.
    - Moving a block requires unstacking all blocks above it first.
    - The arm can only carry one block at a time.

    # Heuristic Initialization
    - Extract the goal conditions to determine the target positions for each
      block.
    - Store the goal parent (block or 'table') for each block mentioned in the
      goal.

    # Step-By-Step Thinking for Computing Heuristic
    1. Parse the current state to determine the current parent (block or 'table')
       and children (blocks on top) for each block.
    2. Check if any block is currently being held.
    3. For each block mentioned in the goal:
        a. If the block is held and not in the goal position, add 1 action
           (putdown or stack).
        b. If the block is not held and not in the goal position, calculate the
           number of blocks above it and add 2 actions per block (unstack and putdown)
           plus 2 actions for the block itself.
    4. Sum all calculated costs to get the heuristic value.
    """

    def __init__(self, task):
        """Initialize the heuristic by extracting goal conditions."""
        self.goal_parent = {}
        for goal in task.goals:
            parts = goal[1:-1].split()
            if parts[0] == 'on':
                block, parent = parts[1], parts[2]
                self.goal_parent[block] = parent
            elif parts[0] == 'on-table':
                block = parts[1]
                self.goal_parent[block] = 'table'
```

```
42
43     def __call__(self, node):
44         """Compute the heuristic value for the given state."""
45         state = node.state
46         current_parent = {}
47         current_children = defaultdict(list)
48         held_block = None
49
50         # Parse current state
51         for fact in state:
52             parts = fact[1:-1].split()
53             if parts[0] == 'on':
54                 child, parent = parts[1], parts[2]
55                 current_parent[child] = parent
56                 current_children[parent].append(child)
57             elif parts[0] == 'on-table':
58                 block = parts[1]
59                 current_parent[block] = 'table'
60             elif parts[0] == 'holding':
61                 held_block = parts[1]
62
63         # Calculate cost
64         cost = 0
65
66         # Check held block
67         if held_block is not None and held_block in self.goal_parent:
68             current_pos = 'held'
69             goal_pos = self.goal_parent[held_block]
70             # If held block is not in goal position, add 1 action
71             if (goal_pos == 'table' and current_pos != 'table') or \
72                (goal_pos != 'table' and current_pos != goal_pos):
73                 cost += 1
74
75         # Process each block in the goal
76         for block in self.goal_parent:
77             if block == held_block:
78                 continue   # Already handled
79
80             current_parent_block = current_parent.get(block, 'table')
81             goal_parent_block = self.goal_parent[block]
82
83             if current_parent_block != goal_parent_block:
84                 # Calculate number of blocks above the current block
85                 def count_above(x):
86                     cnt = 0
87                     stack = [x]
88                     while stack:
89                         current = stack.pop()
90                         for child in current_children.get(current, []):
91                             cnt += 1
92                             stack.append(child)
93                     return cnt
94
95                 above = count_above(block)
96                 cost += 2 * (above + 1)
97
98         return cost
```

## B.2  DeepSeek R1 Heuristic for Spanner

In the *Spanner* domain, an agent must move through a corridor, pick up spanners, and tighten $n$ nuts at the gate, using each picked-up spanner at most once. The agent can move in the direction of the gate but not backwards. Thus, moving without first picking up a required spanner, results in an

unsolvable state. Optimal plans can be computed in polynomial time by simply pick up exactly the first $n$ spanners and then tighten the $n$ nuts at the gate.

However, this is not the strategy the LLM uses to implement the heuristic function. The heuristic greedily assigns to each loose nut (in a fixed order) the closest spanner still available. If the spanner has already been picked up, the cost of tightening the nut is the distance from the agent to the nut location plus 1. If not, then the cost is the distance from the agent to the location of the spanner plus the distance from the location of the spanner to the location of the nut plus 2. Each spanner can be used at most once, so if a nut has no assigned spanner, the heuristic adds a large number to the cost of the state. The heuristic performs a breadth-first search during the initialization phase to compute the shortest path between all locations. This can also overestimate the optimal plan length. Arguably, the LLM could have created a simpler heuristic if the implicit assumptions of the domain were explicit: the PDDL domain allows for arbitrary connections between locations, but all instances assume a one-way corridor.

```python
from fnmatch import fnmatch
from collections import defaultdict, deque
from heuristics.heuristic_base import Heuristic

class spanner10Heuristic(Heuristic):
    """
    A domain-dependent heuristic for the Spanner domain.

    # Summary
    Estimates the number of actions required to tighten all loose nuts by
      considering the minimal path to collect usable spanners and reach each nut's
      location.

    # Assumptions
    - The man (bob) can carry multiple spanners, but each spanner can be used only
      once.
    - The shortest path between locations is precomputed using static link
      information.
    - The man's name is 'bob' as per the problem examples.

    # Heuristic Initialization
    - Extracts static link facts to build a directed graph of locations.
    - Precomputes shortest paths between all pairs of locations using BFS.

    # Step-By-Step Thinking for Computing Heuristic
    1. Identify the man's current location.
    2. Collect all loose nuts and their locations.
    3. Identify all usable spanners (carried or on the ground).
    4. For each loose nut, compute the minimal cost to tighten it using the best
      available spanner.
    5. Assign spanners to nuts greedily, marking used spanners to avoid reuse.
    6. Sum the minimal costs for all nuts to get the heuristic value.
    """

    def __init__(self, task):
        self.goals = task.goals
        self.static_links = defaultdict(list)
        for fact in task.static:
            parts = fact[1:-1].split()
            if parts[0] == 'link':
                start, end = parts[1], parts[2]
                self.static_links[start].append(end)
        # Precompute shortest paths between all locations
        self.shortest_paths = {}
        all_locations = set(self.static_links.keys())
        for ends in self.static_links.values():
            all_locations.update(ends)
        all_locations = list(all_locations)
        for loc in all_locations:
```

```
45                  distances = {loc: 0}
46                  queue = deque([loc])
47                  while queue:
48                      current = queue.popleft()
49                      for neighbor in self.static_links.get(current, []):
50                          if neighbor not in distances:
51                              distances[neighbor] = distances[current] + 1
52                              queue.append(neighbor)
53              self.shortest_paths[loc] = distances
54
55      def __call__(self, node):
56          state = node.state
57          # Find man's location (assumed to be 'bob')
58          man_location = None
59          for fact in state:
60              if fact.startswith('(at bob '):
61                  parts = fact[1:-1].split()
62                  man_location = parts[2]
63                  break
64          if not man_location:
65              return float('inf')  # Invalid state
66
67          # Collect loose nuts and their locations
68          loose_nuts = []
69          nut_locations = {}
70          for fact in state:
71              if fact.startswith('(loose '):
72                  parts = fact[1:-1].split()
73                  loose_nuts.append(parts[1])
74              elif fact.startswith('(at ') and 'nut' in fact:
75                  parts = fact[1:-1].split()
76                  obj, loc = parts[1], parts[2]
77                  nut_locations[obj] = loc
78
79          # Collect usable spanners and their locations
80          usable_spanners = []
81          carried_spanners = []
82          spanner_locations = {}
83          for fact in state:
84              if fact.startswith('(usable '):
85                  parts = fact[1:-1].split()
86                  usable_spanners.append(parts[1])
87              if fact.startswith('(carrying bob '):
88                  parts = fact[1:-1].split()
89                  carried_spanners.append(parts[2])
90              if fact.startswith('(at ') and 'spanner' in fact:
91                  parts = fact[1:-1].split()
92                  spanner_locations[parts[1]] = parts[2]
93
94          # Prepare available spanners (carried or on ground)
95          available_spanners = []
96          for spanner in usable_spanners:
97              if spanner in carried_spanners:
98                  available_spanners.append((spanner, man_location, True))
99              elif spanner in spanner_locations:
100                 available_spanners.append((spanner, spanner_locations[spanner],
    ↪  False))
101
102         # Assign spanners to nuts greedily
103         total_cost = 0
104         used_spanners = set()
105         for nut in loose_nuts:
106             nut_loc = nut_locations.get(nut)
107             if not nut_loc:
108                 continue  # Skip if nut location not found
```

```
109
110             min_cost = float('inf')
111             best_spanner = None
112             for spanner_info in available_spanners:
113                 spanner, s_loc, is_carried = spanner_info
114                 if spanner in used_spanners:
115                     continue
116
117                 if is_carried:
118                     distance = self.shortest_paths[man_location].get(nut_loc,
                        ↪  float('inf'))
119                     cost = distance + 1
120                 else:
121                     d1 = self.shortest_paths[man_location].get(s_loc,
                        ↪  float('inf'))
122                     d2 = self.shortest_paths[s_loc].get(nut_loc, float('inf'))
123                     cost = d1 + 1 + d2 + 1 if d1 != float('inf') and d2 !=
                        ↪  float('inf') else float('inf')
124
125                 if cost < min_cost:
126                     min_cost = cost
127                     best_spanner = spanner
128
129             if best_spanner is not None:
130                 total_cost += min_cost
131                 used_spanners.add(best_spanner)
132             else:
133                 total_cost += 1000000   # Penalize for missing spanner
134
135         return total_cost
```

## B.3   Childsnack

In *Childsnack*, one must prepare and deliver sandwiches to children, some of whom are allergic to gluten. Ingredients are stored at the kitchen and consumed when a sandwich is made. If both ingredients are gluten-free, the resulting sandwich is gluten-free. Sandwiches must be placed on trays at the kitchen, which can be moved between locations, while the children wait at specific locations. The objective is to serve every child by producing sandwiches that respect their allergies.

In the initialization step, the generated heuristic counts the total numbers of allergic and non-allergic children. Then, it counts the unserved allergic and non-allergic children and the number of available gluten-free and regular sandwiches. Next, it estimates the cost to produce and place the missing sandwiches and the cost to move trays to each waiting location. The heuristic value is the sum of all these costs. Because it ignores tray reuse and that gluten-free sandwiches can be served to non-allergic children, it can overestimate the optimal plan length.

## B.4   Floortile

In the *Floortile* domain, robots must paint a grid of tiles with specific colors. Robots may move only onto clear tiles, and moving onto a tile or painting it makes that tile "not clear". Each robot holds a single color and can change to any available color. Painting is performed from an adjacent tile above or below a tile while holding the target color. The objective is to paint all required tiles without blocking access to unpainted tiles that need painting.

The DeepSeek R1 heuristic identifies all required unpainted tiles and their target colors. For each such tile, it calculates the Manhattan distance to every robot and selects the closest ones as candidates. The cost for that tile is estimated as this minimum distance, plus one if none of the candidate robots hold the required color, and another one for the paint action. The total heuristic value is the sum of these costs over all these tiles. This heuristic can overestimate the optimal plan length because it considers tiles independently. Furthermore, by ignoring the blocking constraints that arise from painted tiles, it fails to identify dead-end states.

## B.5 Miconic

The *Miconic* domain models an elevator transporting passengers between floors. This domain is *2-approximable* [37].

The selected heuristic first builds an undirected graph of the floors and precomputes all-pairs shortest paths using breadth-first search. Then, for each unserved passenger, it estimates the remaining cost. If a passenger is already on board, the cost is the distance from the floor of the elevator to its destination plus one (for the depart action). If a passenger is waiting, the cost is the distance from the floor of the elevator to its origin, plus the distance from its origin to its destination, plus two (for the board and depart actions). The total heuristic value is the sum of these costs. Because it ignores that the elevator could serve multiple passengers at once, this heuristic can overestimate the optimal plan length.

## B.6 Rovers

In the *Rovers* domain, a team of rovers explores the surface of a planet to collect data and communicate it back to a lander. Data can be soil or rock samples, or images. Rovers have specific equipment for each task. To collect a sample, a rover must travel to a waypoint, have the right equipment, and an empty storage unit. To take an image, a rover with a camera must first calibrate it at a specific location and then move to a waypoint from which the objective is visible.

During initialization, the selected heuristic processes static information, such as waypoint visibility, builds a traversal graph for each rover, and records their equipment. When evaluating a state, the heuristic iterates through each unachieved goal and estimates the cost to achieve it. If the data has already been collected, the cost is the shortest path for that rover to a waypoint visible from the lander, plus one for the communication action. If the data has not been collected, the cost is estimated by finding an equipped rover that can achieve the goal with minimum cost. This cost includes moving to the goal location, performing the collection or imaging action, moving to a communication waypoint, and communicating. For images, the cost of calibration is also included. Distances are computed on-the-fly using a breadth-first search. The total heuristic value is the sum of these costs over all unachieved goals. This heuristic can also overestimate the optimal plan length.

## B.7 Sokoban

*Sokoban* is a classic **PSPACE**-complete problem [15] where an agent must push boxes to specific goal locations within a grid. The agent can move between adjacent empty locations. To push a box, the agent moves to an adjacent location occupied by a box, and the box is pushed to the next location in the same direction, which must be clear. Since the agent can only push boxes, never pull them, Sokoban has dead-end states.

The heuristic first precomputes all-pairs shortest paths between locations using a breadth-first search. When evaluating a state, it sums the shortest-path distances from each box to its goal location. To this sum, it adds the shortest-path distance from the agent to the closest box not yet at its goal. This heuristic cannot overestimate the optimal plan length, but it ignores push constraints and box interactions and thus fails to identify dead-end states.

## B.8 Transport

In the *Transport* domain, vehicles must deliver packages between locations connected by a road network. Vehicles can move between connected locations, pick up packages, and drop them. The key constraint is that each vehicle has a limited capacity. Picking up a package consumes and dropping it frees one unit of capacity. The objective is to transport all packages to their specified goal locations.

The selected heuristic first precomputes all-pairs shortest paths between locations using a breadth-first search on the road network. When evaluating a state, it iterates through each package not yet at its goal. If a package is already in a vehicle, the cost is the shortest-path distance from the current location of the vehicle to the goal of the package, plus one for the drop action. If the package is at a location, the heuristic greedily assigns it to a vehicle that minimizes the cost, which is calculated as the sum of the travel distance of the vehicle to the package, the travel distance of the package from its current location to its goal, and two actions for pick-up and drop. The total heuristic value is the sum

of these costs for all packages. This heuristic can overestimate the optimal plan length because it does not account for one vehicle delivering multiple packages.

## C  End-to-End Plan Generation Prompt: Blocksworld

Below we show our prompt used for end-to-end plan generation for the smallest task in the test set of the Blocksworld domain. The parts of the prompt that change for different tasks and domains are the name of the domain, the `domain-file`, and `instance-file-1`. As before, to reduce the number of pages, we do not display the entire domain and instance files, but just the beginning of each. We also show only the first few actions of each example plan.

```
1    <problem-description>
2    You are a highly-skilled professor in AI planning searching a plan for a PDDL task
     ↪   from the domain <domain>blocksworld</domain>. You will be given the PDDL
     ↪   domain and the PDDL instance, and you need to return the plan in the format
     ↪   shown below. Next, you will receive a sequence of examples for your task and
     ↪   finally the definition of the blocksworld domain.
3    </problem-description>
4
5    This is the PDDL domain file of the blocksworld domain:
6    <domain-file>
7    (define (domain blocksworld)
8    [...]
9    </domain-file>
10
11   This is the PDDL instance file, for which you need to find a plan:
12   <instance-file-1>
13   (define (problem blocksworld-01)
14   [...]
15   </instance-file-1>
16
17   This is the PDDL domain file of another domain, called Gripper, which serves as an
     ↪   example:
18   <gripper-domain-file>
19   (define (domain gripper-strips)
20   [...]
21   </gripper-domain-file>
22
23   This is an example of an instance file from the Gripper domain:
24   <gripper-instance-file-example>
25   (define (problem strips-gripper-x-20)
26   [...]
27   </gripper-instance-file-example>
28
29   This is a plan for the Gripper instance above:
30   <plan-gripper>
31   (pick ball9 rooma right)
32   (move rooma roomb)
33   (drop ball9 roomb right)
34   (move roomb rooma)
35   [...]
36   </plan-gripper>
37
38   This is the PDDL domain file of another domain, called Logistics, to serve as a
     ↪   second example:
39   <logistics-domain-file>
40   (define (domain logistics-strips)
41   [...]
42   </logistics-domain-file>
43
44   This is an example of an instance file from the Logistics domain:
45   <logistics-instance-file-example>
46   (define (problem strips-log-y-5)
```

```
47   [...]
48   </logistics-instance-file-example>
49
50   This is a plan for the Logistics instance above:
51   <plan-logistics>
52   (load-truck package2 truck12 city5-2)
53   (drive-truck truck12 city5-2 city5-1 city5)
54   [...]
55   </plan-logistics>
56
57   Provide only the plan for the given instance. Here is a checklist to help you with
     ↪  your task:
58   1) The plan must be in the same format as the examples above.
59   2) Use the tags "<plan>...</plan>" around the plan.
60   3) The actions in the plan must be from the set of actions in the domain
     ↪  blocksworld, that is, they must use the same name and same number of
     ↪  parameters as one of the action schemas.
61   4) The plan must be valid, that is, each action must be applicable in the state it
     ↪  is applied in and the plan must end in a goal state.
```

## D   Ablation Study: Simplified Prompt Instructions for Blocksworld

Below we show the simplified instruction component used for the Blocksworld domain within our ablation study. This simplified version replaces the standard detailed instructions, while other parts of the prompt are retained. Only the modified instructions are shown below.

```
1   <problem-description>
2   Create a Python domain-dependent heuristic function for the PDDL domain
    ↪  <domain>blocksworld</domain>. The heuristic function you create will be used
    ↪  to guide a greedy best-first search to solve instances from this domain. The
    ↪  name of the heuristic should be blocksworldHeuristic. Next, you will receive a
    ↪  sequence of file contents to help you with your task and to show you the
    ↪  definition of the blocksworld domain.
3   </problem-description>
```

## E   Runtime Comparison of $h^{\mathrm{FF}}$ in Pyperplan and Fast Downward

Table 5 reports state expansions per second for the $h^{\mathrm{FF}}$ heuristic with GBFS in the Python planner (Pyperplan) and the C++ planner (Fast Downward) on the subset of tasks solved by both. The "Pyperplan" and "Fast Downward" columns report the total number of state expansions divided by the total search time (in seconds) taken to solve the tasks in each domain. This does not include preprocessing time (e.g., for grounding). The "Performance Increase" column is "Fast Downward" divided by "Pyperplan".

Table 5: Expansions per second for GBFS with $h^{\mathrm{FF}}$ in Pyperplan and Fast Downward.

| Domain (# Tasks) | Pyperplan | Fast Downward | Performance Increase |
|---|---|---|---|
| Blocksworld (24) | 111.96 | 8 559.94 | 76.46 |
| Childsnack (17) | 893.26 | 15 890.78 | 17.79 |
| Floortile (10) | 2 758.31 | 87 681.02 | 31.79 |
| Miconic (74) | 1.24 | 829.57 | 669.00 |
| Rovers (27) | 109.04 | 28 710.18 | 263.30 |
| Sokoban (31) | 149.45 | 20 059.83 | 134.22 |
| Spanner (30) | 1 375.35 | 3 255.00 | 2.37 |
| Transport (29) | 3.11 | 305.49 | 98.23 |

