# OpenReview forum: "Classical Planning with LLM-Generated Heuristics: Challenging the State of the Art with Python Code"
_NeurIPS.cc/2025/Conference — NeurIPS 2025 poster_

### Official Review · Reviewer_BKh1 · 2025-06-15

**Clarity:** 3
**Significance:** 2
**Originality:** 2
**Rating:** 4
**Confidence:** 3

**Summary:**

The paper proposes using LLMs to generate domain-dependent heuristic functions in Python and integrating them into the Pyperplan planner for PDDL domains. This pipeline outperforms pure LLM planning that does not use PDDL. Unlike prior work, which prompts the LLM to generate a plan or develops complicated pipelines for LLM-based planning, this paper takes a PDDL description as input and outputs a heuristic that can be directly integrated into an off-the-shelf planner. The results indicate that LLM-generated heuristics can solve more planning tasks than domain-independent heuristics while being competitive with learning algorithms for domain-dependent planning. The LLM-generated heuristics are generally efficient to compute and can even be more informative than state-of-the-art heuristics, resulting in fewer node expansions.

**Questions:**

- Could you run obfuscation experiments, where the domains, predicates, actions, etc., are completely renamed? This would help understand the drivers of performance and whether the LLM is fully utilizing the given PDDL specification to generate heuristics or if the LLM's behavior is similar to parametric retrieval based on the domain name. A small performance degradation could indicate that LLMs can generate heuristics for novel PDDL domains they have not encountered before.
- Could you add more detailed reporting of the quantitative results (e.g., the runtime of heuristic functions in C++/Python, number of expansions, and plan length)? To make the results fully comparable and eliminate the impact of differences in frameworks, re-implementing a set of heuristics in Fast Downward would also be beneficial. A subset of the best R1-generated heuristics would already improve the paper.
- An additional interesting experiment would be to fully adhere to the competition rules (32GB of RAM, no GPU, select suitable model & quantization) and see how far LLMs running on CPUs can be pushed. This would result in fully comparable performance metrics and help assess how close LLMs really are to the state of the art in this domain, considering that "Challenging the State of the Art" appears in the paper's title. This is a nice-to-have suggestion, not a requirement.

**Ethical Concerns:**

["NO or VERY MINOR ethics concerns only"]

**Final Justification:**

The authors' rebuttal addresses the main concerns of generalization to other PDDL domains, lack of detail in the description of the results, and experiments with smaller models. I think the paper can be accepted if these changes are incorporated. I believe the main remaining weaknesses are the somewhat incremental nature of the paper (using LLMs to generate heuristics has been proposed before), the somewhat narrow application area of PDDL planning, and the relative lack of generalizable insights to other problem areas.

**Limitations:**

- There are no significant ethics concerns or potential negative social impact. Limitations are discussed interspersed throughout the text. The authors could point out that this method relies on PDDL. While the proposed method is significantly superior to purely LLM-based planning when the problem is easily expressible in PDDL, LLM-based planning does not have this dependency, which makes it a more general approach.

**Quality:**

2

**Strengths And Weaknesses:**

Strengths
- The paper tackles a relevant research question: How can LLMs be integrated into classical planning? The quality of the heuristic function is essential for efficient planning, so applying an LLM to improve it makes sense.
- The paper is well-written and easy to follow.
- The method is simple compared to complex LLM planning pipelines, which is a significant strength. The results are promising and consistent across domains. The computational efficiency gains over end-to-end LLM planning are significant, especially as the generated heuristics can be re-used for multiple tasks.
- The difference between this work and prior work is discussed in detail.

Weaknesses
- The method uses LLMs to create domain-specific heuristics for well-known planning domains. Some of the domains are very well-known, and, to my knowledge, all of them have been published before. It is unclear whether the LLM is genuinely synthesizing novel heuristics based on the PDDL description or if it is simply performing retrieval, ensembling, and/or adaptation of known solutions for known problems. Prompting the LLM for 25 candidates could essentially be hill-climbing on memorized fragments.
- A Python-based planner outperforming a C++-based Fast Downward is an impressive result. However, the paper could provide more details, such as average/median number of node expansions and plan length, to understand the significance of these results. Figure 2 gives some answers but is insufficient alone.
- While the issue with memory limit for the training phase is documented (Lines 138-139), it is still a concern in terms of fairness. Furthermore, the IPC 2023 competition rules did not allow the use of GPUs. Considering the size of DeepSeek models and the lack of public details on Gemini models, only the Distill Qwen 14B is somewhat comparable to the baselines in terms of memory usage during inference. This model is outperformed by the domain-independent heuristic h^{FF}. On the other hand, recent work [1] has noted that successfully scaling up GPU compute in these tasks is a non-trivial problem, with neural network-based methods struggling to outperform classical ML methods. From this perspective, this paper suggests a promising solution.
- In addition to FunSearch and Ling et al., which the paper cites, there is other recent work [2-5] doing LLM-based heuristic generation for non-PDDL tasks. In addition to LLM-based sampling of heuristics proposed by this paper, some of these papers include reflective evolution and search that utilizes diversity. On the other hand, the authors acknowledge the feedback loop as a direction for future work, and the paper achieves strong results in PDDL domains even without it.

[1] Chen, D. Z., Trevizan, F., & Thiébaux, S. (2024). Return to tradition: learning reliable heuristics with classical machine learning. In Proceedings of the International Conference on Automated Planning and Scheduling (Vol. 34, pp. 68-76).

[2] Dat, P. V. T., Doan, L., & Binh, H. T. T. (2025). Hsevo: Elevating automatic heuristic design with diversity-driven harmony search and genetic algorithm using llms. In Proceedings of the AAAI Conference on Artificial Intelligence (Vol. 39, No. 25, pp. 26931-26938).

[3] Ye, H., Wang, J., Cao, Z., Berto, F., Hua, C., Kim, H., ... & Song, G. (2024). Reevo: Large language models as hyper-heuristics with reflective evolution. NeurIPS 2024

[4] Liu, F., Tong, X., Yuan, M., Lin, X., Luo, F., Wang, Z., ... & Zhang, Q. (2024). Evolution of heuristics: Towards efficient automatic algorithm design using large language model. ICML 2024

[5] Zheng, Z., Xie, Z., Wang, Z., & Hooi, B. (2025). Monte Carlo Tree Search for Comprehensive Exploration in LLM-Based Automatic Heuristic Design. arXiv preprint arXiv:2501.08603.

---

> ### Author Rebuttal · Authors · 2025-07-31
>
> Thank you for the suggestions and for your careful read of the paper! We address your concerns and present the requested experiments below.
>
> > Could you run obfuscation experiments, where the domains, predicates, actions, etc., are completely renamed? This would help understand the drivers of performance and whether the LLM is fully utilizing the given PDDL specification to generate heuristics or if the LLM's behavior is similar to parametric retrieval based on the domain name. A small performance degradation could indicate that LLMs can generate heuristics for novel PDDL domains they have not encountered before.
>
> To address the concern of parametric retrieval based on the domain name, we ran three additional experiments designed to test whether the LLMs are retrieving memorized heuristics or reasoning about the provided PDDL specification.
>
> 1. Domain Modification (Spanner): We modified the Spanner domain so that spanners are not consumed after use. If the LLM had merely memorized a known heuristic for Spanner, it would fail to account for this change. However, the generated heuristic for the modified domain correctly adapted to the new semantics. The best generated heuristic is indeed *perfect* for this modified domain, and thus  produces optimal plan costs for all states. This suggests the LLM reasoned about the given domain rather than retrieved a memorized solution.
>
> 2. Predicate Obfuscation (Blocksworld): We created an obfuscated version of the Blocksworld domain where all domain, predicates, actions, etc... are completely replaced with random strings as done by Valmeekam et al. 2023. We translated the Blocksworld tasks used in the paper and ran an experiment with the exact same setup as in the paper. The results are shown in the table below:
>
> | domain                 | Fast Downward FF | Pyperplan R1 |
> |------------------------|------------------|--------------|
> | Blocksworld            | 27               | 66           |
> | Obfuscated Blocksworld | 28               | 40           |
>
> As shown in the table, the performance of the classical planner (Fast Downward with FF) remained stable (modulo noise), as expected. The performance of the R1-generated heuristic degraded but still outperformed the FF baseline. This indicates that while semantic cues in predicate names are helpful, the LLM is still capable of reasoning about the logical structure of the domain. We note that this experiment is a highly adversarial setting for LLMs, because they are trained to take into account the semantics of the tokens and not operate on random strings.
>
> 3. Novel Domain (Rod-Rings): We designed a completely new domain, Rod-Rings, which has never been made available online nor fed to an LLM. It features sticks with stacks of rings and a single "held" ring. Two moves are allowed: (a) place the held ring at the bottom of a stick, swapping it with the top ring, or (b) place the held ring on top of a stick, swapping it with the bottom ring. The goal is to arrange specific rings on specific sticks in a defined order.
>
> To ensure the domain remains novel, we used an OpenAI model (O3) via their API, which guarantees that API data is not used for model training nor stored. To be able to compare to the results of the paper, we also provide results for O3-generated heuristic on the domains used in the paper.
>
> | domain                       | Fast Downward FF | Pyperplan O3 |
> |------------------------------|------------------|--------------|
> | Rod-Rings                    | 59               | 58           |
> | Paper Domains (IPC Learning) | 295              | 354          |
>
> On this new domain, the O3-generated heuristic guiding a GBFS solves 58 tasks, almost as many as Fast Downward guided by FF which solves 59 tasks. This suggest that the LLM reasons about this unseen domain.
> The results for the IPC Learning domains for the O3-generated heuristics are inferior to the R1-generated heuristics which solve 373 tasks.
>
> Together, these experiments -- modifying an existing domain, obfuscating one domain and testing on a novel domain -- indicate that the success our method is not due to parametric retrieval based on the domain name but to the LLMs' ability to generate domain-dependent heuristic functions by reasoning about the logical structure of the domain.
>
> - Valmeekam et al.: On the Planning Abilities of Large Language Models - A Critical Investigation. NeurIPS 2023.
>
> > Could you add more detailed reporting of the quantitative results (e.g., the runtime of heuristic functions in C++/Python, number of expansions, and plan length)? To make the results fully comparable and eliminate the impact of differences in frameworks, re-implementing a set of heuristics in Fast Downward would also be beneficial. A subset of the best R1-generated heuristics would already improve the paper.
>
> Yes, we will add a detailed report with the requested information in the appendix of the camera-ready version. Below you can find the a table with the number of expansions per second for the FF heuristic in Pyperplan and Fast Downward for the subset of tasks solved by both planners. The table shows the difference in performance between the two planners using the same search algorithm (GBFS) and the same heuristic (FF).
>
> | Domain (Tasks)        | Python   | C++       | Performance Increase |
> |-----------------------|----------|-----------|----------------------|
> | Blocksworld (24)      | 6964.21  | 142621.78 | 20.48                |
> | Childsnack (17)       | 14495.45 | 190385.84 | 13.13                |
> | Floortile (10)        | 27486.89 | 229067.58 | 8.33                 |
> | Miconic (74)          | 4878.33  | 37715.63  | 7.73                 |
> | Rovers (27)           | 3410.42  | 63064.41  | 18.49                |
> | Sokoban (31)          | 1350.62  | 37400.46  | 27.70                |
> | Spanner (30)          | 25312.26 | 44137.59  | 1.74                 |
> | Transport (29)        | 1162.04  | 8559.27   | 7.37                 |
>
> As we can see, the C++ is significantly faster in general. In 7 out of the 8 domains, the C++ implementation is at least 7x faster and sometimes even 20x faster.
>
> We used an LLM to directly translate the R1-generated heuristics for the two domains with the lowest coverage, Childsnack and Floortile, into C++ code for Fast Downward. The results are shown in the table below:
>
> | Domain      | Pyperplan R1 | Fast Downward R1 |
> |-------------|--------------|------------------|
> | Childsnack  |           22 |               26 |
> | Floortile   |            4 |                5 |
>
> The results for "Fast Downward R1" should be taken with a grain of salt, as the translation from Python to C++ is very direct and not optimized in any way. Also, our method chooses the heuristic (among the 25 generated heuristics) that works best in Pyperplan, not in C++.
>
> > In addition to FunSearch and Ling et al., which the paper cites, there is other recent work doing LLM-based heuristic generation for non-PDDL tasks. In addition to LLM-based sampling of heuristics proposed by this paper, some of these papers include reflective evolution and search that utilizes diversity. On the other hand, the authors acknowledge the feedback loop as a direction for future work, and the paper achieves strong results in PDDL domains even without it
>
> Thank you for the suggestions! We agree that adding to our approach techniques, such as reflective evolution and search that utilize diversity, could bring even further benefits. However, our method shows that even a straightforward method can already be useful to create classical planning heuristics.
>
> > An additional interesting experiment would be to fully adhere to the competition rules (32GB of RAM, no GPU, select suitable model & quantization) and see how far LLMs running on CPUs can be pushed. This would result in fully comparable performance metrics and help assess how close LLMs really are to the state of the art in this domain, considering that "Challenging the State of the Art" appears in the paper's title. This is a nice-to-have suggestion, not a requirement.
>
> We ran a local experiment with Qwen3 4B using a NVIDIA GeForce RTX 4090 with 24GB of VRAM. This is a popular GPU that is commonly used also in personal computers. To generate 25 heuristics sequentially, we need 4h33m on average. (One could generate two heuristics in parallel with the available VRAM, but we stick to a single process.)
>
> The results are as follows:
>
> | Domain       | Qwen3 4B |
> |--------------|----------|
> | Blocksworld  |       27 |
> | Childsnack   |       12 |
> | Floortile    |        3 |
> | Miconic      |       90 |
> | Rovers       |       31 |
> | Sokoban      |       26 |
> | Spanner      |       30 |
> | Transport    |       44 |
> | Total        |      263 |
>
>
> This shows that a "small LLM" can already outperform the FF heuristic in Pyperplan. It has strictly higher coverage in 4 out of the 8 domains, and it ties with FF in another domain.
>
> As LLMs are optimized for GPU used, running an experiment without GPUs is impractical. Moreover, it is not uncommon in the planning literature that methods use GPU during learning phase, and switch to CPU-only during planning. A few recent examples (including Chen et al., which introduce $h^{WLF}\_{GPR}$ that we compared to):
>
> - Dillon Z. Chen, Sylvie Thiébaux, Felipe W. Trevizan: Learning Domain-Independent Heuristics for Grounded and Lifted Planning. AAAI 2024: 20078-20086
> - Dillon Z. Chen, Felipe W. Trevizan, Sylvie Thiébaux: Return to Tradition: Learning Reliable Heuristics with Classical Machine Learning. ICAPS 2024: 68-76
> - Simon Ståhlberg, Blai Bonet, Hector Geffner: Learning Generalized Policies without Supervision Using GNNs. KR 2022
> - Simon Ståhlberg, Blai Bonet, Hector Geffner: Learning General Policies with Policy Gradient Methods. KR 2023: 647-657
>
> We could include this result as a new column in our Table 1, if the reviewers find it useful.

---

> > ### Comment · Reviewer_BKh1 · 2025-08-04
> >
> > Thank you for your response, the clarifications & new results address my concerns effectively. I have no further questions.

---

### Official Review · Reviewer_MPk1 · 2025-06-23

**Clarity:** 3
**Significance:** 4
**Originality:** 3
**Rating:** 5
**Confidence:** 5

**Summary:**

This paper shows how LLMs can be used to generate domain-dependent heuristics which can be integrated off-the-shelf into Pyperplan, a python based planner developed with educational purposes in mind, not performance. The methodology prompts an LLM providing the PDDL representation of the target domain, with the smallerst and largest instance. The prompt is complemented with two examples of planning domains, their models, their solution in python, their static domain and state representation in Pyperplan along with a checklist of common errors. A sample of heuristics are generated, n=25, and the best performing one over the training instances from the learning competition in the International Planning Competition (IPC)  is selected given the agile score of the IPC, with a five minutes timeout.

The experimental results strongly suggest that this approach to derive a heuristic function is competitive with state-of-the-art domain-independent heuristics, as well as other learning approaches to develop domain-dependent heuristic based on Gaussian models. Overall, the generated heuristic yields better coverage with similar expansion rates and plan length, meaning that the increase in coverage doesn't come with a cost of computation time, as a function of expanded nodes, and plan quality.

Overall, it doesn't only outperform planners developed for performance considerations and well known domain-independent heuristics, it also shows how it substantially increases the performance of LLMs without reasoning capabilities.

**Questions:**

- What is the speed factor difference when you try to solve a task with a highly engineered planner in C++ vs a planner written in python with educational purposes? This result can help the reader understand the impact of table 2. It could be measured by expansion time running breadth first search, or running the same LLM heuristic and greedy best first search, one written in python, the other one adapted by the authors in C++.

- Can you clarify further on the issues raised above with respect to understanding what general strategies the heuristic is following, and how different it is from work published in the literature?

**Ethical Concerns:**

["NO or VERY MINOR ethics concerns only"]

**Final Justification:**

The authors addressed my questions properly. I think they also addressed criticisms from other reviewers. This paper provides valuable insights and the experimental results are strong.

**Quality:**

3

**Strengths And Weaknesses:**

# Quality

The paper develops a sound methodology following the style of papers aiming to understand how to prompt an LLM for coding purposes. It evaluates in depth how to assess the value of the approach by comparing the generated pipeline with existing approaches in the planning literature. Several metrics are used, baselines are provided using different LLMs, planning codebases, domain-independent heurstics and learnt heuristics, and the approach seems to be easily reproducible following the appendix.

On the flip side, a minor comment is that I missed having more details on the computation time to create the heuristic, as well as computation time to evaluate the heuristic. It is only mentioned in text with as a few hours. See question below.

I also missed a deeper analysis of the strategies used by the LLM to develop their heuristics. The paragraph before section 5 is really insightful, but I was left wanting to know more, which per se it is a good outcome of a paper, but can also be a negative one. In trying to understand and relate the heuristics to theory, or to general strategies in planning, as a reviewer I was left wondering if all heuristics would follow the structure of the heuristics developed by correlation complexity [1], using weighted linear combinations of state variables, or other general strategies in the literature similar to relaxations, decomposition, etc. Having a deeper dive on what the heuristic "looks like" would make the paper more insightful.

No mention was added with respect of testing problems that have already been used for training the LLM. The paper cited below contains heuristics which are very similar to the one produced by the LLM in Blocksworld, as well as well known strategies to solve blocksworld suboptimally. Understanding how novel the new heuristics are (in terms of general strategies for problem solving) could address this issue, as well as devising new domains.

[1] Jendrik Seipp, Florian Pommerening, Gabriele Röger, and Malte Helmert. 2016. Correlation complexity of classical planning domains. In Proceedings of the Twenty-Fifth International Joint Conference on Artificial Intelligence (IJCAI'16). AAAI Press, 3242–3250.

# Clarity
The experimental design follows well accepted performance metrics by the Planning community, which is essential to convey the results. The text is well written and easy to follow.

Minor suggestions:
- move footnote 4 to the main text, it's an important observation.
- Table 3, clarify number of failed heuristics out of how many? The table could also be more informative if shown as a figure.

# Significance

In the context of how to make LLMs to plan, this represents a promising approach to lift their performance closer to state-of-the-art symbolic planners, and at the same time, it provides a way forward to integrate LLMs within symbolic planning systems. This has been proved to be a very difficult task, with all the literature so far showing how far LLMs are in their planning capabilities when faced with planning problems specified in PDDL.

The results in this paper clearly motivate an active research direction developing this methodology further for other planning formalisms, improving LLMs planning ability and the integrations of LLMs in the development of symbolic planners.

# Originality

This approach is not novel, i.e. programming a heuristic as noted by the authors has been tried before, but it has never been shown to perform so well on satisficing classical planning, a problem that so far has been proved to be extremely challenging for LLMs.

---

> ### Author Rebuttal · Authors · 2025-07-31
>
> Thank you for your comments and suggestions! We clarify some of your points below.
>
> > What is the speed factor difference when you try to solve a task with a highly engineered planner in C++ vs a planner written in python with educational purposes? This result can help the reader understand the impact of table 2. It could be measured by expansion time running breadth first search, or running the same LLM heuristic and greedy best first search, one written in python, the other one adapted by the authors in C++.
>
> Below we show a table with the number of expansions per second for the FF heuristic in Pyperplan (implemented in Python) and Fast Downward (implemented in C++) for the subset of tasks solved by both planners (shown in parentheses). The table shows the difference in performance between the two planners using the same search algorithm (GBFS) and the same heuristic (FF).
>
> | Domain (Tasks)        | Python   | C++       | Performance Increase |
> |-----------------------|----------|-----------|----------------------|
> | Blocksworld (24)      | 6964.21  | 142621.78 | 20.48                |
> | Childsnack (17)       | 14495.45 | 190385.84 | 13.13                |
> | Floortile (10)        | 27486.89 | 229067.58 | 8.33                 |
> | Miconic (74)          | 4878.33  | 37715.63  | 7.73                 |
> | Rovers (27)           | 3410.42  | 63064.41  | 18.49                |
> | Sokoban (31)          | 1350.62  | 37400.46  | 27.70                |
> | Spanner (30)          | 25312.26 | 44137.59  | 1.74                 |
> | Transport (29)        | 1162.04  | 8559.27   | 7.37                 |
>
> Please also note because of the different *grounding techniques* used by Pyperplan and Fast Downward there are tasks where Pyperplan fails in the grounding phase and cannot start the search while Fast Downward succeeds and can start the search.
>
> Pyperplan fails in 162 tasks during grounding, while Fast Downward fails only in 65. These are tasks that we cannot solve, regardless how good the heuristics are. So there is 2.5x more tasks that Pyperplan cannot possibly solve compared to Fast Downward.
>
> (Reviewer BKh1 asked a similar question, so it might be useful to read our answers to their questions as well.)
>
> > Can you clarify further on the issues raised above with respect to understanding what general strategies the heuristic is following, and how different it is from work published in the literature?
>
> We will use the additional content page of the camera-ready version to add the description of the general strategies used by LLM-generated heuristics. In addition to Spanner and Blocksworld, we will also add a brief description of the main ideas of the R1-generated heuristics that where selected in our training phase.
>
> Below we briefly summarize the general strategy used by these selected heuristics.
>
> These LLM-generated heuristics *do not use general strategies common in the planning literature* such as linear combinations of state variables, correlation complexity, relaxations, or standard decomposition methods.
>
> In most generated heuristics the general strategy is a domain-dependent decomposition of the problem. The heuristic identifies the remaining subgoals (packages to be delivered or tiles to be painted) and sums the estimated costs to achieve each one independently. This often involves calculating movement costs via shortest-path algorithms (precomputed or calculated on-demand) and adding costs for other necessary actions (picking up an object or painting a tile). The heuristics then offset the final value to take into account identified interactions between these subgoals.
>
> >  I missed having more details on the computation time to create the heuristic, as well as computation time to evaluate the heuristic.
>
> The time to generate the heuristic depends on the limitations of the LLM API used. For the case of Gemini 2.0 Flash, we ask for three heuristics per minute, thus to generate the 200 heuristics we need 67 minutes. However, the limit of the API is 15 calls per minute, thus we could reduce the time to 13 minutes. In the case of DeepSeek models there is no limit for the number of calls, thus we can generate the 200 heuristics simultaneously. We generate the R1 heuristics in a longer period to monitor the process, but it could be done in parallel.
>
> Regarding the time to evaluate the heuristics. We have chosen to use all training tasks to select the best heuristic to obtain the simplest approach possible. However, our method allows using a smaller set of tasks to select the best heuristic if the computational resources are limited.
>
> We ran two additional experiments to analyze this further. We used two different subsets of tasks from the training set to select the best R1-generated heuristic: six tasks distributed over the training set (including the smallest and the largest task), and two tasks (only the smallest and the largest task). We ran the experiment with the exact same setup as in the paper. The results are shown in the table below:
>
> | Training Subset                     | Tasks Solved |
> |-------------------------------------|--------------|
> | Smallest, Largest and 4 Random Tasks| 374          |
> | Smallest and Largest Task           | 361          |
> | All Training Tasks                  | 373          |
>
> As shown in the table, using only six tasks we achieve performance similar to using all training tasks. In this case, the worst case time to evaluate the 25 heuristics per domain over the six tasks with a time limit of five minutes would be 25 * 6 * 5 = 750 minutes (twelve hours and thirty minutes), which is far less than the 24 hours available per domain during the IPC Learning Track 2023. We could further reduce this time by using only two tasks which would result in four hours and ten minutes.
>
> > Table 3, clarify number of failed heuristics out of how many? The table could also be more informative if shown as a figure.
>
> Thank you for the suggestion! The number of failed heuristics is out of 200 (25 * 8). We will clarify this in the paper.
>
> > No mention was added with respect of testing problems that have already been used for training the LLM. The paper cited below contains heuristics which are very similar to the one produced by the LLM in Blocksworld, as well as well known strategies to solve blocksworld suboptimally.
>
> To address the concern of problems used for the training the LLM, we ran three additional experiments designed to test whether the LLMs are retrieving memorized heuristics or reasoning about the provided PDDL domain.
>
> 1. Domain Modification (Spanner): We modified the Spanner domain so that spanners are not consumed after use. If the LLM had merely memorized a known heuristic for Spanner, it would fail to account for this change. However, the generated heuristic for the modified domain correctly adapted to the new semantics. The best generated heuristic is indeed *perfect* for this modified domain, and thus  produces optimal plan costs for all states. This suggests the LLM reasoned about the given domain rather than retrieved a memorized solution.
>
> 2. Predicate Obfuscation (Blocksworld): We created an obfuscated version of the Blocksworld domain where all components names were replaced with random strings as done by Valmeekam et al. 2023. We translated the Blocksworld tasks used in the paper and ran an experiment with the exact same setup as in the paper. The results are shown in the table below:
>
> | domain                 | Fast Downward FF | Pyperplan R1 |
> |------------------------|------------------|--------------|
> | Blocksworld            | 27               | 66           |
> | Obfuscated Blocksworld | 28               | 40           |
>
> As shown in the table, the performance of the classical planner (Fast Downward with FF) remained stable (modulo noise), as expected. The performance of the R1-generated heuristic degraded but still outperformed the FF baseline. This indicates that while semantic cues in predicate names are helpful, the LLM is still capable of reasoning about the logical structure of the domain. We note that this experiment is a highly adversarial setting for LLMs, because they are trained to take into account the semantics of the tokens and not operate on random strings.
>
> 3. Novel Domain (Rod-Rings): We designed a new domain, Rod-Rings, which has never been made available online nor fed to an LLM. It features sticks with stacks of rings and a single "held" ring. Two moves are allowed: (a) place the held ring at the bottom of a stick, swapping it with the top ring, or (b) place the held ring on top of a stick, swapping it with the bottom ring. The goal is to arrange specific rings on specific sticks in a defined order.
>
> To ensure the domain remains novel, we used an OpenAI model (O3) via their API, which guarantees that API data is not used for model training nor stored. To be able to compare to the results of the paper, we also provide results for O3-generated heuristic on the domains used in the paper.
>
> | domain                       | Fast Downward FF | Pyperplan O3 |
> |------------------------------|------------------|--------------|
> | Rod-Rings                    | 59               | 58           |
> | Paper Domains (IPC Learning) | 295              | 354          |
>
> On this new domain, the O3-generated heuristic guiding a GBFS solves 58 tasks, almost as many as Fast Downward guided by FF which solves 59 tasks. This suggest that the LLM reasons about this unseen domain.
> The results for the other domains for the O3-generated heuristics are inferior to the R1-generated heuristics which solve 373 tasks.
>
> Together, these experiments indicate that the success our method is not due to heuristics memorized during training but to the LLMs' ability to generate domain-dependent heuristic functions by reasoning about the logical structure of the domain.
>
> - Valmeekam et al.: On the Planning Abilities of Large Language Models - A Critical Investigation. NeurIPS 2023.

---

> > ### Comment · Reviewer_MPk1 · 2025-08-03
> > **Discussion**
> >
> > I thank the authors for their response. They addressed all my questions with additional experiments that clarify further the contribution being made.
> >
> > The structure of the heuristics extracted still remind me to the ones proposed in the correlation complexity paper, where one ends up with a weighted combination of features. in the case of the LLMs, the weights are extracted from measures such as shortest path, and the selection of features is mostly constrained to the goals. This is just an observation, it does not affect my valuation of the paper.

---

### Official Review · Reviewer_VEfu · 2025-07-01

**Clarity:** 3
**Significance:** 1
**Originality:** 1
**Rating:** 2
**Confidence:** 4

**Summary:**

The paper investigates the usage of LLM for solving classical planning tasks defined in PDDL. Instead of generating an end-to-end plan, the authors propose generating domain-dependent heuristics that are used in a greedy best-first search. The method consists of choosing a best-of-n heuristic by the score on the training set. The experiments include 8 classic planning environments. Results demonstrate that generated heuristics outperform domain-independent baselines and show competitive performance to the domain-dependent one.

**Questions:**

The paper requires redirecting, as the general problem that the authors are trying to solve is unclear. Is it proposing the new SOTA LLM-based method for classical planning? If so, the results should be SOTA, and perhaps the paper is a better fit for a planning conference. Is it a new hybrid method for planning, applicable to the general natural text planning problems? Then, such applications and results should be shown and compared to the suitable baselines.

**Ethical Concerns:**

["NO or VERY MINOR ethics concerns only"]

**Final Justification:**

The proposed method shows results **comparable** to SOTA in PDDL-formalized environments, and there could be more effort put into developing (e.g., fine-tuning) the method to outperform the baseline, considering the focus of the paper only on the PDDL-like tasks. If the method is still considered an alternative to general end-to-end planning (as one of the baselines), then the FLOPs-controlled experiments on **non-PDDL environments** (e.g., the proposed scheme of problem formalization -> heuristic generation -> solution) should be conducted.
I appreciate the author's efforts in rebuttal; however, I am inclined to keep my score due to the aforementioned concerns.

**Limitations:**

Yes

**Quality:**

2

**Strengths And Weaknesses:**

Strengths:
 - The paper is well-written and easy to follow, with enough background information provided.

Weaknesses:
 - The PDDL environments are often used to benchmark the LLM's planning abilities in general, which is why the plans are generated in full, end-to-end. While the models are known to fail on larger, out-of-distribution tasks, the key here is that the model can be applied to general-case planning for problems not defined in PDDL (e.g., agentic and robotics tasks). Although in the related work section, the authors discuss the possibility of incorporating their heuristic generation method into the pipeline where natural text problems are translated into PDDL, they do not attempt it, making the comparison unfair.
 - Overall, the method is a very simplistic BoN, with no modifications made to make it specific for the planning problems. Since the results are comparable to the only one provided domain-dependent baseline (371 $h^{WLF}_{GPR}$ vs. 373 $h^{R1}$), I would expect either a much greater margin, which would indeed demonstrate surprising abilities of the model to generate domain-specific heuristics, or some modifications/training specific for planning.
- The statement of the baselines being state-of-the-art is questionable, since IPC 2023 shows the winners of GOFAI & HUZAR, which are not used in the comparison.
- The possible contamination issue is not addressed. Though unlikely, there is a possibility of such heuristics present in the training set of R1 & Gemini (which were released after the IPC 2023).
 - Ablation study shows large deviations, which makes the statistical significance questionable.

---

> ### Author Rebuttal · Authors · 2025-07-30
>
> Thank you for your review and for identifying some issues with the presentation!
>
> > The paper requires redirecting, as the general problem that the authors are trying to solve is unclear.
>
> We respectfully disagree with the reviewer. The problem we address is the well-documented limitation of LLMs to reliably solve problems that require reasoning (Valmeekam et al., 2023; Stechly et al., 2024; Shojaee et al., 2025).
>
> We propose a novel approach to this well-known problem: instead of using LLMs end-to-end, we use them to generate domain-dependent heuristic functions that guide a search algorithm. This approach can be understood as using LLM for program synthesis.
> As shown in Table 1, our method dramatically improves performance. For example, for the non-reasoning model Gemini 2.0, our approach increases the number of solved tasks from 19 to 296.
>
> > Is it proposing the new SOTA LLM-based method for classical planning? If so, the results should be SOTA [...] The statement of the baselines being state-of-the-art is questionable, since IPC 2023 shows the winners of GOFAI & HUZAR, which are not used in the comparison.
>
> This seems to be a misunderstanding. Our results are indeed state-of-the-art for the problem we are solving.
>
> Our approach generates heuristic functions to guide a GBFS. Consequently, we compare our method against several domain-independent heuristics available in the Fast Downward planner and against the best learning-based method for domain-dependent heuristics, $h^{WLF}\_{GPR}$. Notably, $h^{WLF}\_{GPR}$ is far superior to any other heuristic in the IPC 2023 Learning Track. All these baseline heuristics also guide a GBFS, ensuring a fair comparison.
>
> GOFAI and HUZAR are not heuristic functions. They are complete planning systems that incorporate several advanced techniques that make them incomparable to a single heuristic function guiding a simple GBFS.
>
> Investigating how to use our techniques as one of the components within such a planning system is an interesting direction for future work.
>
> We agree that some of our statements regarding state-of-the-art performance (e.g., on lines 240 and 260) could be clearer. We will reword them to be more precise in the final version. Thank you for pointing this out!
>
> > Since the results are comparable to the only one provided domain-dependent baseline (371 $h^{WLF}\_{GPR}$ vs. 373 $h^{R1}$)...
>
> We could compare our method to other domain-dependent baselines if the reviewers agree. However, to the best of our knowledge, $h^{WLF}\_{GPR}$ is currently the strongest domain-dependent heuristic for classical planning.
> $h^{WLF}\_{GPR}$ is the culmination of decades of research into how to learn domain-dependent heuristics. Our results show that LLMs can often generate domain-dependent heuristics that are competitive with $h^{WLF}\_{GPR}$ running inside the C++ planner Fast Downward, even if we execute them in an educational Python planner.
>
> > .. the paper is a better fit for a planning conference.
>
> We respectfully disagree and believe our paper is an excellent fit for NeurIPS. The NeurIPS community has shown a significant and growing interest in topics central to our work, such as planning, reasoning, and program synthesis using LLMs.
>
> To illustrate this, we provide a non-exhaustive list of recent papers on LLMs for planning that were published at NeurIPS or raised a lot of interest in the ML community:
>
> - Zhao et al.: Improving Large Language Model Planning with Action Sequence Similarity. ICLR 2025.
>
> - Katz et al.: Thought of Search: Planning with Language Models Through The Lens of Efficiency. NeurIPS 2024.
>
> - Stechly et al.: Chain of Thoughtlessness? An Analysis of CoT in Planning. NeurIPS 2024.
>
> - Valmeekam et al.: On the Planning Abilities of Large Language Models - A Critical Investigation. NeurIPS 2023.
>
> - Guan et al.: Leveraging Pre-trained Large Language Models to Construct and Utilize World Models for Model-based Task Planning. NeurIPS 2023.
>
> - Shojaee et al.: The Illusion of Thinking: Understanding the Strengths and Limitations of Reasoning Models via the Lens of Problem Complexity. 2506.06941 Apple 2025
>
> > The PDDL environments are often used to benchmark the LLM's planning abilities in general [...] the authors discuss the possibility of incorporating their heuristic generation method into the pipeline where natural text problems are translated into PDDL, they do not attempt it, making the comparison unfair.
>
> Research on improving the capabilities of LLMs to solve problems that require reasoning is ongoing.
>
> We believe that building a complete pipeline from natural language to a correct plan should involve multiple steps. The research community is tackling these challenges through various modular approaches, such as the ones focusing on translating the problem into a formal representation (Gestrin et al., 2024; Katz et al., 2024; Tantakoun et al., 2025).
>
> Our work focuses specifically on the plan generation component, assuming a formal PDDL model is already available.
>
> While an end-to-end system is a long-term goal, we believe that focusing on and improving individual components is a crucial step in advancing the field.
>
> - Tantakoun et al.: LLMs as Planning Formalizers: A Survey for Leveraging Large Language Models to Construct Automated Planning Models. ACL (Findings) 2025.
>
> - Gestrin et al.: NL2Plan: Robust LLM-Driven Planning from Minimal Text Descriptions. 2405.04215 2024.
>
> > Overall, the method is a very simplistic BoN, with no modifications made to make it specific for the planning problems.
>
> Simplicity is an advantage and not a problem. Our method shows that even a straightforward method can already be useful to create classical planning heuristics. We agree that adding layers of modifications tailored to planning can bring even further benefit.
>
> > Ablation study shows large deviations, which makes the statistical significance questionable.
>
> The large deviation in the ablation study is an expected and desired outcome of our method. Our goal is not to generate heuristics that are good on average, but to ensure that we have a diverse pool of heuristic to select from. The high variance is a direct consequence of sampling with a high temperature to maximize diversity. To obtain more consistent results, one could simply use a lower temperature. However, this is not desirable. We ran preliminary experiments with lower temperatures resulting in lower deviation but also lower coverage.
>
> > The possible contamination issue is not addressed. Though unlikely, there is a possibility of such heuristics present in the training set of R1 & Gemini.
>
> To address the concern of potential data contamination, we ran three additional experiments designed to test whether the LLMs are retrieving memorized heuristics or reasoning about the provided PDDL domain.
>
> 1. Domain Modification (Spanner): We modified the Spanner domain so that spanners are not consumed after use. If the LLM had merely memorized a known heuristic for Spanner, it would fail to account for this change. However, the generated heuristic for the modified domain correctly adapted to the new semantics. The best generated heuristic is indeed *perfect* for this modified domain, and thus produces optimal plan costs for all states. This suggests the LLM reasoned about the given domain rather than retrieved a memorized solution.
>
> 2. Obfuscation (Blocksworld): We created an obfuscated version of the Blocksworld domain where all component names were replaced with random strings as done by Valmeekam et al. We translated the Blocksworld tasks used in the paper and ran an experiment with the exact same setup as in the paper. The results are shown in the table below:
>
> | domain                 | Fast Downward FF | Pyperplan R1 |
> |------------------------|------------------|--------------|
> | Blocksworld            | 27               | 66           |
> | Obfuscated Blocksworld | 28               | 40           |
>
> The performance of Fast Downward with FF remained stable (modulo noise), as expected. The performance of the R1-generated heuristic degraded but still outperformed the FF baseline. This indicates that while semantic cues in predicate names are helpful, the LLM is still capable of reasoning about the logical structure of the domain. We note that this experiment is a highly adversarial setting for LLMs, because they are trained to take into account the semantics of the tokens and not operate on random strings.
>
> 3. Novel Domain (Rod-Rings): We designed a new domain, Rod-Rings, which has never been made available online nor fed to an LLM. It features sticks with stacks of rings and a single "held" ring. Two moves are allowed: (a) place the held ring at the bottom of a stick, swapping it with the top ring, or (b) place the held ring on top of a stick, swapping it with the bottom ring. The goal is to arrange specific rings on specific sticks in a defined order.
>
> To ensure the domain remains novel, we used an OpenAI model (O3) via their API, which guarantees that API data is not used for model training nor stored. We also provide results for O3-generated heuristic on the domains used in the paper.
>
> | domain                       | Fast Downward FF | Pyperplan O3 |
> |------------------------------|------------------|--------------|
> | Rod-Rings                    | 59               | 58           |
> | Paper Domains (IPC Learning) | 295              | 354          |
>
> On this new domain, the O3-generated heuristic guiding a GBFS solves 58 tasks, almost as many as Fast Downward guided by FF which solves 59 tasks. This suggest that the LLM reasons about this unseen domain.
> The results for the other domains for the O3-generated heuristics are inferior to the R1-generated heuristics which solve 373 tasks.
>
> Together, these experiments indicate that the success our method is not due to contamination but to the LLMs' ability to generate domain-dependent heuristic functions by reasoning about the logical structure of the domain.

---

> > ### Author Response · Authors · 2025-08-05
> >
> > Dear Reviewer VEfu,
> >
> > Thank you again for your detailed review and suggestions.
> >
> > We would like to know if you still have any remaining questions or concerns about our submission, so that we can clarify them during the discussion phase. We believe that we addressed all your comments and answered all the questions, but we would very much prefer to hear your opinion.
> >
> > As the discussion phase is halfway through, it would be great if you could let us know whether your opinion has changed after the rebuttal, as we only have a few days left for the discussion.
> >
> > Thank you for your time and contributions to the review process.

---

> > > ### Comment · Reviewer_VEfu · 2025-08-05
> > >
> > > Thank you for your clarifications.
> > >
> > > > instead of using LLMs end-to-end, we use them to generate domain-dependent heuristic functions that guide a search algorithm.
> > >
> > > As mentioned in the review and the papers you provided, the classical PDDL-defined planning domains are typically used to evaluate the planning abilities of the LLMs. However, the end-to-end approach **is applicable** to planning and reasoning in any domain, not just PDDL-formalized, whereas the proposed approach of generating a heuristic relies heavily on the provided PDDL description being available and correct. I want to point out that this makes the comparison unfair and shifts the focus of the paper towards solving classical PDDL tasks. Acknowledging the recent developments in formalization, the experiments on non-formalized domains are necessary (for example, by going problem -> formalization -> heuristics generation -> solution) for a truly novel contribution to the LLM-based planning field. The fair comparison to end-to-end baselines in a FLOP- and time-controlled experiment would resolve the issue.
> > >
> > > If the paper focuses exclusively on PDDL tasks, then it’s important to report scores from domain-specific baselines. Additionally, demonstrating a more substantial performance gap (e.g., 371 vs. 373) would likely require further methodological improvements—for example, training, prompt optimization, or heuristic integration—which would not only meet the novelty requirement but also constitute a valuable contribution to ongoing research in the area.
> > >
> > > > To address the concern of potential data contamination, we ran three additional experiments designed to test whether the LLMs are retrieving memorized heuristics or reasoning about the provided PDDL domain.
> > >
> > > I agree that the possibility of data contamination is addressed.

---

> > > > ### Author Response · Authors · 2025-08-06
> > > >
> > > > Thank you for carefully reading our review and for your explanation.
> > > >
> > > > > However, the end-to-end approach is applicable to planning and reasoning in any domain, not just PDDL-formalized, whereas the proposed approach of generating a heuristic relies heavily on the provided PDDL description being available and correct. I want to point out that this makes the comparison unfair and shifts the focus of the paper towards solving classical PDDL tasks.
> > > >
> > > > We now understand your point, and we agree with you. Our goal with the end-to-end comparison was not to claim that our method can completely replace the end-to-end approach. We included this comparison just as a reference. We will emphasize this in the camera-ready copy, and we will rephrase the paper to remove any ambiguity in this regard.
> > > >
> > > > > The fair comparison to end-to-end baselines in a FLOP- and time-controlled experiment would resolve the issue.
> > > >
> > > > We compare below the results for end-to-end and heuristic generation in terms of FLOPs and time. We report the results for DeepSeek R1.
> > > >
> > > > **FLOPs**:
> > > >
> > > > From our experimental data, we can estimate the FLOPs for each method. We use the optimistic estimate that DeepSeek R1 uses 74 billion FLOPS per tokens. This is a common and optimistic back-of-the-napkin estimate based on the number of active parameters of R1.
> > > >
> > > > For end-to-end, we have the following data:
> > > >
> > > > - Avg. tokens per task: 25762.01 tokens
> > > > - Avg. tokens per domain (90 tasks): 2318580.84 tokens
> > > >
> > > > - Total FLOPs per domain: ~141 quadrillion
> > > >
> > > > For our approach, we have the following:
> > > >
> > > > - Avg per heuristic: 26329.06 tokens
> > > > - Avg per domain (25 heuristics): 658226.53 tokens
> > > >
> > > > - Total FLOPs per domain: ~48 quadrillion
> > > >
> > > > Compared to the ~141 quadrillion FLOPs used by the end-to-end approach, this is a large reduction. Note also that if the number of tasks in the test set increased, the total number of FLOPs required for the end-to-end planning approach would increase, while for our method would remain intact.
> > > >
> > > > For the search part of our approach (i.e., running Pyperplan), there are close to zero FLOPs, as the planner has almost no floating point operation. To account for that, we can consider the total number of instructions instead. The highest number of instructions necessary to solve a task in our benchmark was 1.2 trillion, which is significantly lower than the quantity of FLOPs used by the LLMs per task/heuristic.
> > > >
> > > > **Time**:
> > > >
> > > > The longest time needed for end-to-end plan generation was 19m36s (an instance in the Transport domain). If we use 19m36s as a cutoff, we would have the following results:
> > > >
> > > > | Coverage at 19m36s | End-to-End | Heuristic Generation |
> > > > |--------------------|------------|----------------------|
> > > > | Blocksworld        | 17         | 64                   |
> > > > | Childsnack         | 40         | 22                   |
> > > > | Floortile          | 0          | 3                    |
> > > > | Miconic            | 24         | 90                   |
> > > > | Rovers             | 10         | 31                   |
> > > > | Sokoban            | 8          | 30                   |
> > > > | Spanner            | 47         | 68                   |
> > > > | Transport          | 28         | 60                   |
> > > > | **Total**              | 174        | 368                  |
> > > >
> > > > Our method solves only 5 fewer tasks than if we allow for the 30m usually given in the IPC.
> > > >
> > > > > If the paper focuses exclusively on PDDL tasks, then it’s important to report scores from domain-specific baselines.
> > > >
> > > > To the best of our knowledge, $h^{WLF}\_{GPR}$ is the state-of-the-art domain-specific baseline PDDL planning. $h^{WLF}\_{GPR}$ learns one heuristic for each domain, and uses the exactly same set of training data of at most 99 tasks per domain as we did. It first runs an off-the-shelf PDDL planner for up to 30 minutes on each task of the training set to extract a set of optimal plans. Then, it uses these plans to learn a heuristic for the domain.
> > > >
> > > > If you have some additional baseline in mind, could you please give us a pointer so we can include it in the paper?

---

### Official Review · Reviewer_K8zo · 2025-07-01

**Clarity:** 3
**Significance:** 2
**Originality:** 2
**Rating:** 3
**Confidence:** 4

**Summary:**

This paper proposes a novel method to improve classical planning by leveraging large language models (LLMs) to generate domain‐dependent heuristic functions from PDDL domain descriptions. Instead of relying on end-to-end plan generation by the LLM, the method “induces” heuristics in the form of Python code that guide a greedy best-first search. The approach involves prompting the LLM multiple times with a detailed and multi-component prompt (including PDDL domain files, sample tasks, example heuristics, and even a checklist of pitfalls), collecting diverse candidate heuristics, evaluating their performance on a training set of planning tasks, and finally selecting the best performing heuristic to tackle out-of-distribution test tasks. The paper includes extensive experiments comparing the LLM-generated heuristics with traditional domain-independent heuristics (e.g., FF) and state-of-the-art learning-based planners. Ablation studies on prompt components further illustrate the impact of various parts of the prompt. Overall, the work demonstrates that LLM-generated heuristics can significantly enhance planning performance and even overcome performance gaps between Python-based prototypes and C++ implementations in existing planners.

**Questions:**

- How sensitive is the overall performance to changes in the prompt formulation? Could more concise prompts yield similar performance, or is the detailed multi-component prompt essential?
- How do variations in the underlying LLM (e.g., reasoning vs. non-reasoning models or model scale) affect the quality of the generated heuristics? Is there a clear trade-off between computational cost and heuristic performance? The LLMs in the paper are very strong.
- Maybe worth discuss the LLM domain induction works on PDDL for better position the paper.

**Ethical Concerns:**

["NO or VERY MINOR ethics concerns only"]

**Final Justification:**

It’s clear from the rebuttal that the current version of the paper overlooks several important aspects. The next draft should incorporate missing components such as the small model experiments, detailed ablation studies, a discussion of limitations, and potentially a more thorough review of related work.

Overall, the paper would benefit from another round of careful revision and editing to address these gaps.

**Limitations:**

I don't see a separate limitation section. Not clearly discussed in Section 4 as well.

**Quality:**

3

**Strengths And Weaknesses:**

Strengths:
- The paper presents an original idea of using LLMs to “induce” domain-dependent heuristics, circumventing the drawbacks of end-to-end plan generation and manual heuristic design.
- A thorough experimental evaluation is provided, with comparisons against well-known baselines and state-of-the-art methods; the experiments on IPC domains, ablation studies, and runtime comparisons add credibility to the results.
- The paper carefully explains the heuristic generation pipeline, including prompt design, candidate generation, evaluation, and selection. Supplementary material provides detailed prompts and example code, which aids reproducibility and transparency.
- Although demonstrated on classical planning with PDDL, the approach has potential to be extended to other planning or combinatorial task settings.

Weaknesses:
- The method appears highly sensitive to the design and length of the prompt. The appendix shows an extremely long and intricate prompt—raising questions about scalability and whether the approach would still work as effectively with shorter or less detailed instructions.
- The process of selecting the “best” heuristic from a generated set is somewhat heuristic itself, and additional discussion on the sensitivity of this selection mechanism and its potential failure modes would strengthen the work.

---

> ### Author Rebuttal · Authors · 2025-07-30
>
> Thank you for your review and for your suggestions! We answer your questions below.
>
> > How sensitive is the overall performance to changes in the prompt formulation? Could more concise prompts yield similar performance, or is the detailed  multi-component prompt essential?
>
> Our ablation study (page 7, line 245) analyzes the sensitivity of our method to the prompt formulation. As detailed in Table 3, the performance ceiling for the original prompt is much higher than for all cases where components are individually removed, which shows that all components strengthen our method.
>
> At the same time, the ablation study also shows that while all components strengthen our method, most can be removed individually without a significant drop in performance. This suggests that the prompt is quite robust to changes.
>
> > How do variations in the underlying LLM (e.g., reasoning vs. non-reasoning models or model scale) affect the quality of the generated heuristics?
>
> Table 1 shows that reasoning models (DeepSeek R1 and Gemini Thinking) outperform their non-reasoning versions (DeepSeek V3 and Gemini).
>
> However, it is interesting to notice that the gap between their performances is not that large. This is in contrast to the end-to-end plan generation, where the reasoning models are vastly superior to their non-reasoning counterparts. In other words, *our method can reduce the performance gap between reasoning and non-reasoning models, allowing us to use the non-reasoning and cheaper models to obtain a similar performance*.
>
> > Is there a clear trade-off between computational cost and heuristic performance? The LLMs in the paper are very strong.
>
> We ran a local experiment with Qwen3 4B using a NVIDIA GeForce RTX 4090 with 24GB of VRAM. This is a popular GPU that is commonly used in personal computers. To generate 25 heuristics sequentially, we need 4h33m on average. (One could generate two heuristics in parallel with the available VRAM, but we stick to a single process.)
>
> We repeat the exact experiment of the paper with these new heuristics, and the results are as follows:
>
> | Domain       | Qwen3 4B |
> |--------------|----------|
> | Blocksworld  |       27 |
> | Childsnack   |       12 |
> | Floortile    |        3 |
> | Miconic      |       90 |
> | Rovers       |       31 |
> | Sokoban      |       26 |
> | Spanner      |       30 |
> | Transport    |       44 |
> | Total        |      263 |
>
>
> This shows that a "small LLM" can already outperform the FF heuristic in Pyperplan. It has strictly higher coverage in 4 out of the 8 domains, and it ties with FF in another domain.
>
> > Maybe worth discuss the LLM domain induction works on PDDL for better position the paper.
>
> Do you suggest discussing the literature on extracting PDDL domains from natural language? Yes, we're happy to do that. This has the benefit of showing that our method can be used in conjunction with such methods, providing an end-to-end solution for planning on natural language tasks, via PDDL.
>
> > I don't see a separate limitation section. Not clearly discussed in Section 4 as well.
>
> We will add a limitation section in the camera-ready version.
>
> > The process of selecting the “best” heuristic from a generated set is somewhat heuristic itself, and additional discussion on the sensitivity of this selection mechanism and its potential failure modes would strengthen the work.
>
> Thank you for the suggestion. We will add a discussion on the sensitivity of the selection mechanism in the camera-ready version. We already ran two additional experiments to analyze this further. We used two different subsets of tasks from the training set to select the best R1-generated heuristic: six tasks distributed over the training set (including the smallest and the largest task), and two tasks (only the smallest and the largest task). The results are shown in the table below:
>
> | Training Subset                     | Tasks Solved |
> |-------------------------------------|--------------|
> | Smallest, Largest and 4 Random Tasks| 374          |
> | Smallest and Largest Task           | 361          |
> | All Training Tasks                  | 373          |
>
> As shown in the table, the selection mechanism is quite robust to the choice of the training tasks. The Table shows that *using only two tasks* (the smallest and the largest) *still results in a good performance*. This suggests that the selection mechanism is robust to the choice of training tasks. We have chosen to use all training tasks to select the best heuristic to obtain the simplest approach possible. However, our method allows using a smaller set of tasks to select the best heuristic if the computational resources are limited.

---

### Official Review · Reviewer_2xoe · 2025-07-11

**Clarity:** 2
**Significance:** 3
**Originality:** 3
**Rating:** 3
**Confidence:** 3

**Summary:**

This work showed sampling a set of planning heuristic functions can significantly improve the planning capabilities of LLMs. More specifically, for a given planning domain, LLM generates several domain-dependent heuristic functions in the form of Python code, evaluates them on a set of training tasks with a greedy best-first search, and chooses the best one. They showed that the LLM-generated heuristic functions solve more tasks than LLMs prompted to generate plans end-to-end, and this difference is especially pronounced for non-reasoning LLMs.

**Questions:**

please take a look at weakness and questions ablove.

**Ethical Concerns:**

["NO or VERY MINOR ethics concerns only"]

**Final Justification:**

Some of my concerns are discussed in the rebuttal and it also highlight some of the issues I was pointing out. Since the heuristics are generated by LLMs they might not be reliable and also the search over them can be expensive. This need to be studied more thoroughly since the search is part of the planning itself.

**Limitations:**

yes

**Quality:**

2

**Strengths And Weaknesses:**

Strengths:

- The paper uses LLM to generate heuristics for the problem which seems interesting and showing that these heuristics may outperform the existing planner is surprising.

- The paper provide comparison against existing planner

Weaknesses/Questions:

- It seems the LLM itself is not solving the planning problems and it produces the heuristics that still need the solver to find the plan. Also the paper mentioned that these heuristics outperforms existing planners which I don’t understand because if there exists a better heuristic then we should be able to design a better approximate algorithm. For example, for the blocksworld, there is a known 2-approx algorithm. Does that mean the generated heuristic by LLMs does better? If yes, is there any evidence for that?

- The paper mentioned that LLM-generated heuristic functions solve more tasks than LLMs prompted to generate plans end-to-end. But it is not LLM that eventually executes the heuristics. Also I am wondering if the generated heuristics are executable or they still need human check!

- In a more general case, the complexity of classical planning problems varies from polynomial to the Pspace. Does that mean LLMs are able to give us a reasonable approximate solution for np-complete, Pspace problems?

- h^{FF} is referred to in the paper but never explained what the heuristic is. Especially when saying LLM's heuristics outperform state-of-the-art heuristics, such as h^{FF}. There should be some examples and justification. Also there are other heuristics mentioned in the paper which need to be explained for better understanding of LLMs’ heuristics.

- The paper should discuss the details on datasets distribution on size, etc. They only mentioned an example like this: In Blocksworld, for example, the largest training task contains 29 146 blocks, while the largest test task has 488 block

---

> ### Author Rebuttal · Authors · 2025-07-30
>
> Thank you for your review and for your suggestions! We clarify your questions below.
>
> > It seems the LLM itself is not solving the planning problems and it produces the heuristics that still need the solver to find the plan. Also the paper mentioned that these heuristics outperforms existing planners which I don’t understand because if there exists a better heuristic then we should be able to design a better approximate algorithm. For example, for the blocksworld, there is a known 2-approx algorithm. Does that mean the generated heuristic by LLMs does better? If yes, is there any evidence for that?
>
> Regarding the comparison to approximation algorithms, it is important to clarify the research context. The goal in automated planning is to *develop domain-independent planners that work sufficiently well for any domain*. These planners use domain-independent heuristic functions that reason about the general PDDL structure without any built-in knowledge of "Blocksworld" or any other domain.
>
> An approximation algorithm, like the 2-approximation for Blocksworld, is a specialized algorithm for a single domain. Comparing a general planner to a specialized algorithm is not a standard evaluation practice as their objectives are different.
>
> You are correct that in our approach the LLM does not solve the planning tasks directly. In our paper, we show that using LLMs for end-to-end plan generation is less effective (solving 202 tasks) than a planner guided by a domain-independent heuristic function (solving 324 tasks). However, we also show that we can use LLMs to generate domain-dependent heuristic functions that outperform the state-of-the-art domain-independent heuristics (solving 373 tasks).
>
> > The paper mentioned that LLM-generated heuristic functions solve more tasks than LLMs prompted to generate plans end-to-end. But it is not LLM that eventually executes the heuristics.
>
> Thank you for pointing this out!
> When we say that LLM-generated heuristic functions solve more tasks than LLMs prompted to generate plans end-to-end, we mean that *greedy best-first search guided by the LLM-generated heuristic function* solves more tasks. We will clarify this.
>
> > Also I am wondering if the generated heuristics are executable or they still need human check!
>
> There are *no human checks* in our approach. Instead, the process is fully automated.
>
> We prompt the LLM to generate heuristic functions as Python code. These generated functions are then automatically evaluated on a set of training tasks. The best-performing heuristic is automatically selected and the Python code is then directly injected into the planner and executed to solve the unseen test tasks.
>
> > Does that mean LLMs are able to give us a reasonable approximate solution for np-complete, Pspace problems?
>
> LLMs can give us heuristic function that provide reasonable estimates for NP-hard problems, but we still need to run a search algorithm to find the solution. This search can still, in the worst case, expand an exponential number of states.
>
> However, a good heuristic can make the search much more efficient in practice by guiding it toward promising states and pruning large parts of the search space that do not lead to a solution. Our results show that LLM-generated heuristics can be very effective and often outperform state-of-the-art heuristic functions.
>
> > h^{FF} is referred to in the paper but never explained what the heuristic is. Especially when saying LLM's heuristics outperform state-of-the-art heuristics, such as h^{FF}. There should be some examples and justification. Also there are other heuristics mentioned in the paper which need to be explained for better understanding of LLMs’ heuristics.
>
> Thank you for the suggestion! We will use the additional content page of the camera-ready version to describe the domain-independent heuristics mentioned in the paper.
>
> > The paper should discuss the details on datasets distribution on size, etc.
>
> Good idea! We will include additional details about the dataset. Below you can find the size distribution of the the most significant parameters for training and test set tasks:
>
> | domain      | Training size | Testing size |
> |-------------|---------------|--------------|
> | Blocksworld | n ∈ [2, 29]   | n ∈ [5, 488] |
> | Childsnack  | c ∈ [1, 10]   | c ∈ [4, 292] |
> | Floortile   | t ∈ [2, 30]   | t ∈ [12, 980]|
> | Miconic     | p ∈ [1, 10]   | p ∈ [1, 485] |
> | Rovers      | r ∈ [1, 4]    | r ∈ [1, 30]  |
> | Sokoban     | b ∈ [1, 4]    | b ∈ [1, 79]  |
> | Spanner     | s ∈ [1, 10]   | n ∈ [1, 487] |
> | Transport   | v ∈ [1, 7]    | n ∈ [3, 50]  |
>
> Most significant parameter for each domain: n blocks in Blocksworld, c children in Childsnack, t tiles in Floortile, p passengers in Miconic, r rovers in Rovers, b boxes in Sokoban, s spanners in Spanner, v vehicles in Transport.

---

> > ### Author Response · Authors · 2025-08-05
> >
> > Dear Reviewer 2xoe,
> >
> > Thank you again for your detailed review and suggestions.
> >
> > We would like to know if you still have any remaining questions or concerns about our submission, so that we can clarify them during the discussion phase. We believe that we addressed all your comments and answered all the questions, but we would very much prefer to hear your opinion.
> >
> > As the discussion phase is halfway through, it would be great if you could let us know whether your opinion has changed after the rebuttal, as we only have a few days left for the discussion.
> >
> > Thank you for your time and contributions to the review process.

---

### Note · Authors · 2025-08-13

Dear AC and Reviewers,

We thank all reviewers for their valuable feedback, which has helped us significantly strengthen our paper and better present our core contributions.  Reviewers BKh1 and MPk1 confirmed that our rebuttal and new experiments effectively addressed their concerns.

Our work introduces a new paradigm to improve the planning capabilities of LLMs by synthesizing heuristic function code. This guarantees valid plans and dramatically improves performance, increasing solved tasks from 243 to 373 using the heuristics generated with DeepSeek R1, for example. We believe our paper presents a simple, powerful, and validated method that significantly improves the planning capabilities of LLMs.

A key point of discussion concerned our SOTA comparison. We emphasize that comparing our heuristic functions (in GBFS) against other SOTA heuristic functions in GBFS is the correct and fair evaluation methodology. The result is highly significant not only because of the difference in coverage but also because it reveals that a simple LLM-based synthesis approach allows the discovery of heuristic functions that are superior to domain-independent heuristic functions resulting from decades of research and are competitive to the state-of-the-art domain-specific heuristic function baseline for PDDL planning. This surprising result challenges long-held assumptions about heuristic design.

To address the reviewer's concerns, we performed new experiments on novel, obfuscated, and modified domains that provide strong evidence against data contamination. Reviewers MPk1 and VEfu acknowledged that this addressed their concerns, validating that the LLM synthesizes a heuristic from the PDDL specification.

In the final version, we will incorporate these new results and address all the raised points. We will refine our presentation and claims to improve clarity and precision. In particular, we will rephrase our SOTA comparisons to ensure the full context is always present. Following Reviewer VEfu's feedback, we will clarify that our end-to-end LLM comparison serves as a reference point, not as a claim to replace the end-to-end paradigm. Furthermore, we will add a dedicated discussion of our method's limitations.

Thank you for your consideration.

---

### Decision · Program_Chairs · 2025-09-17

**Decision:**

Accept (poster)

**Comment:**

# Meta Review

Please provide detailed and convincing justifications for your recommendation, including but not limited to
(a) Summarize the scientific claims and findings of the paper based on your own reading and characterizations from the reviewers.
(b) What are the strengths of the paper?
(c) What are the weaknesses of the paper? What might be missing in the submission?
(d) Provide the most important reasons for your decision to accept/reject. For spotlights or orals explain why the paper stands out (other than by high scores or popularity trends).
(e) Summarize the discussion and changes during the rebuttal period. What were the points raised by the reviewers? How were each of these points addressed by the authors? How did you weigh in each point in your final decision? (Do not mention reviewer names. Use their anon ids instead.)

# Summary

The paper proposes an approach of leveraging LLMs for classical planning approaches. Specifically, the proposed approach is to
1. Use LLMs to generate multiple domain-specific heuristics function as a python code
2. Then use the training set tasks with greedy bfs to compare the generated heuristics and select the best one
3. Finally, use the selected heuristics to tackle out-of-distribution test tasks.

The experiments compare their heuristics  with traditional domain-independent heuristics as well as SOTA domain-specific heuristics. They also compare the proposed planning approach to LLM-based end-to-end planning. Their experiments show signigicant improvement in coverage of tasks with the LLM-generated heuristics.

# Strengths

- **Novelty**:  The paper situates itself well within the recent surge of LLM‑based planning research and is well motivated to contribute improvements to a key component---heuristics generation---in planning pipeline.
- **Comprehensive Evalations**: The paper reports comparison of the proposed LLM-generated heuristics with domain-dependent and domain-independent heuristics as well as it compares with ned-to-end LLM based approach. The ablation study provides sensitivity analysis for each component of the prompt. Finally, the discussion on generated heuristics provide good insight on the LLMs ability to be used for the such tasks.
- **Transparency & Reproducibility**:  Supplementary material provides detailed prompts and example code, which aids reproducibility and transparency. (quoting Reviewer K8zo)
- **Reliability**: Since the proposed approach uses LLMs for a well-defined component, the produced code can be verified and evaluated in a controlled manner. Promoting reliability of the proposed solution.


# Weakness

- **Limitation on natural-language planning problems**: One major concern is that the proposed approach is not evaluated on planning problems that are expressed in natural language. The experiments on the pipeline `NL -> PDDL -> Heuristic Generation -> Solve with Planner` would have addressed that. But authors mention it as future work. (Reviewer VEfu)
- **Limited Details on existing heuristics**: The paper lacks explanation of existing heuristics used in the baseline (Reviewer 2xoe)



# Reviewer concerns


- **Relevance to NeurIPS**: One major concern raised (by reviewer VEfu) is the fit for NeurIPS. As the paper proposes an approach specifically for planning problems described in PDDL, the paper might be a better fit for a planning conference. I tend to disagree.
    1. The submission to NeurIPS are not restricted by the formal representation used by a paper. The call for papers in NeurIPS mention:
> We invite submissions presenting new and original research on topics including but not limited to the following:[..] Reinforcement learning (e.g., decision and control, planning, hierarchical RL, robotics).
    2. There is enough interest in planning and the PDDL representation at NeurIPS itself, as illustrated by authors.
- **Prompt Engineering**: Reviewer K8zo raises that the prompt used to generate the heuristics is long, has multiple components (including PDDL instance and domain files, two examples from another domains, and a checklist for instructions). However, their ablation study analyzes the sensitivity to prompts. Additionally, the baseline domain-dependent heuristics approaches get similar information as that mentioned in the prompt, so comparison is fair as well.
- **Unfair end-to-end planning baseline comparison**: The paper proposed a heuristics generation approach that is used for planning problems described in PDDL. They compare it with SOTA domain-specific heuristics as well as end-to-end planning approaches. Reviewer VEfu highlights that comparison to end-to-end planning approaches is unfair for coverage and a comparison that is controlled for FLOPS and time would resolve the issue. The authors provide convincing evidence that even when controlled for FLOPS the heuristics generation out-performs end-to-end. Although this was trivial given heuristics generation is domain-specific and end-to-end is problem specific. Besides, End-to-end planning is one of the baselines. The paper also compares against domain-independent and domain-dependent heuristics
- **Small Gap from the best existing heuristics**: Reviewer VEfu highlights that the presented approach does not have significant improvement over $h^{WLF}_{GPR}$ and hence more exploration in terms of approaches should be done.
    1. I believe the difference in the coverage is still significant given that $h^{WLF}_{GPR}$ was run in C++ and the LLM generated heuristic was generated
    2. While I agree that much more can be done to show more substantial gap, but the novelty of the approach is not undermined by lack of exploration of dufferent approaches..
- **Experiments with Smaller Models**: (Reviewer K8zo) The experiments uses large, strong models for heuristics generation. It is not clear if the approach would work with smaller models. The authors rebut this by providing results with smaller Qwen model (4B).
- **Retrieval from memory**: Reviewer  VEfu and MPk1 raised the concerns that the evaluated domains might already be part of pre-training data in LLM. The author rebuttal addresses this concern by providing experiment results with domain modification, obfuscation and introducing a novel domain. All of which makes the paper stronger.

# Recommendation Justification

The paper makes a solid contribution to using LLMs for planning besides extracting work-model or translating NL to PDDL, which has been explored by existing literatures. The thorough empirical envaluations and provided supplementary invokes confidence in the proposed approach.  The remaining concerns from the reviewer do not take away from the contribution of the paper; as I reflect in the reviewer concerns section above.